# A spatial perturbation framework to validate implantation of the epileptogenic zone

Kassem Jaber [1,2], Tamir Avigdor [1,3], Daniel Mansilla[4], Alyssa Ho[1,5], John Thomas[1,2], Chifaou Abdallah[1,3], Stephan Chabardes [6], Jeff Hall[7], Lorella Minotti[6], Philippe Kahane [6], Christophe Grova[3,8,9], Jean Gotman[9] & Birgit Frauscher [1,2,5] ✉

Stereo-electroencephalography (SEEG) is the gold standard to delineate surgical targets in focal drug-resistant epilepsy. SEEG uses electrodes placed directly into the brain to identify the seizure-onset zone (SOZ). However, its major constraint is limited brain coverage, potentially leading to misidentification of the 'true' SOZ. Here, we propose a framework to assess adequate SEEG sampling by coupling epileptic biomarkers with their spatial distribution and measuring the system's response to a perturbation of this coupling. We demonstrate that the system's response is strongest in well-sampled patients when virtually removing the measured SOZ. We then introduce the spatial perturbation map, a tool that enables qualitative assessment of the implantation coverage. Probability modelling reveals a higher likelihood of well-implanted SOZs in seizure-free patients or non-seizure free patients with incomplete SOZ resections, compared to non-seizure-free patients with complete resections. This highlights the framework's value in sparing patients from unsuccessful surgeries resulting from poor SEEG coverage.

The success of epilepsy surgery relies heavily on the precision of presurgical planning, which typically involves the use of clinical semiology, electroencephalography (EEG) and magnetic resonance imaging (MRI)[1] to identify the epileptogenic zone (EZ). However, in complex cases, these tools may not be sufficient, and stereotactic EEG (SEEG) electrodes need to be inserted into the brain to obtain a better spatial resolution[2]. Bancaud and Talairach defined the EZ as the network or region of earliest seizure organization and propagation[3]. While SEEG can be extremely useful, it is limited by the constraint of implanting a finite number of electrodes; indeed, a typical SEEG exploration with $192 \pm 54$ contacts covers only about 5–10% of brain volume[4–6].

The seizure-onset zone (SOZ) is an electrographic marker defined as the channels exhibiting the earliest changes during a seizure in the EEG[7]. In clinical practice, the SOZ is used as a proxy marker of the EZ during presurgical planning. Even with careful identification of the potential epileptic focus, only half of the patients selected for surgery become seizure-free after the procedure[8]. One of the primary reasons for surgical failure is the inability to properly sample the 'true' SOZ[9–11] resulting in misidentification of the EZ. This inaccuracy can be attributed to several factors, such as ambiguous hypotheses derived from phase 1 presurgical evaluation, as well as inaccuracies involved in the stereotactic implantation of the SEEG electrodes.

[1]Analytical Neurophysiology Lab, Montreal Neurological Institute and Hospital, McGill University, Montréal, QC, Canada. [2]Department of Biomedical Engineering, Duke Pratt School of Engineering, Durham, NC, USA. [3]Multimodal Functional Imaging Lab, Biomedical Engineering Department, McGill University, Montréal, QC, Canada. [4]Neurophysiology Unit, Institute of Neurosurgery Dr. Asenjo, Santiago, Chile. [5]Department of Neurology, Duke University Medical Center, Durham, NC, USA. [6]Grenoble Institute Neurosciences, Inserm, U1216, CHU Grenoble Alpes, Université Grenoble Alpes, Grenoble, France. [7]Department of Neurology and Neurosurgery, Montreal Neurological Institute and Hospital, McGill University, Montréal, QC, Canada. [8]Multimodal Functional Imaging Lab, School of Health, Department of Physics, Concordia University, Montréal, QC, Canada. [9]Montreal Neurological Institute, McGill University, Montréal, QC, Canada. ✉e-mail: birgit.frauscher@duke.edu

Until now, the literature has mainly focused on developing biomarkers for localizing the EZ without considering whether the electrode locations will even allow for the detection of the EZ; this however is a prerequisite for accurate localization[12]. We have identified a few studies that have attempted to address this issue. The first study developed an approach to estimate brain activity from 'missing channels'[13]. While this approach can refine the precision of epileptic focus localization and minimize functional deficits during resection, it is limited when trying to evaluate whether the implantation missed the focus, given that SEEG contacts are only sensitive to neural activities up to a distance of 10 mm[6]. A second study proposed a new source localization methodology informed by SEEG recordings[14]. Using this approach, it is possible to detect interictal epileptiform discharges (IEDs) in areas not sampled by SEEG using magnetoencephalography (MEG) recordings. However, this method requires an extra set of MEG recordings, which are not available in many epilepsy centers. In addition, missing IEDs does not necessarily mean that the implantation is poor, given that the irritative zone is typically larger than the EZ[15]. A third study applied electrical source imaging on propagated spikes recorded from SEEG[16]. It showed a predominant outward information flow from the spike onset in seizure-free patients. This might suggest that good EZ coverage is required to observe this flow, but this was not assessed in this work. Other studies have applied electrical source imaging methods on SEEG data to improve the localization of the EZ[17], or predict surgical outcomes[18,19]. They have shown that localizing IEDs or seizures recorded by the SEEG can localize far-field activity not sampled by the SEEG, and using this information, predict surgical outcomes. However, they did not develop a model to score the adequate sampling of the EZ by a given electrode configuration. Therefore, there is a need to develop a simple model based on SEEG alone to evaluate whether a given implantation scheme has adequately sampled the epileptic focus.

Here, we propose a framework which uses interictal biomarkers to measure the success of a given SEEG electrode configuration in sampling the EZ as part of presurgical planning. The use of interictal biomarkers for the development of the framework can enable one to evaluate the SEEG configuration without the need to record seizures. Therefore, we opted to only consider interictal biomarkers in this study since our ultimate goal was to perform a seizure-independent evaluation of the SEEG configuration. More specifically, we considered IEDs with preceding gamma activity (30–100 Hz; IED-$\gamma$), as this interictal marker has been shown to be highly specific to the EZ[11,20]. This study will consider a well-sampled patient as one whose electrode configuration is sufficient to identify and resect the presumed EZ leading to seizure freedom (Engel IA outcome). Two outcomes may arise as a result of implementing our proposed analysis. First, if the analysis indicates that the SEEG implantation completely missed the EZ, the epileptologists would need to re-evaluate the initial hypothesis on the EZ location originating from the non-invasive investigation and no surgery should be pursued. Second, if the EZ was only partially sampled, additional SEEG electrodes can be inserted in a second step to improve the hypothesis on the EZ, leading to better postsurgical outcome. The fundamental concept behind this framework is to transform epileptic features at the channel-level into a spatial system. Ultimately, our framework may be a valuable tool in the presurgical evaluation stage, as it may effectively prevent unsuccessful but not complication-free surgeries.

## Results

### Patient characteristics

The study involved two cohorts: the cohort from the Montreal Neurological Institute (MNI) (50 patients; 17 Engel IA, 33 Engel IIB+) and the cohort from the Grenoble Alpes University Hospital Center (CHUGA) (26 patients; 18 Engel IA, 8 Engel IIB+). We excluded Engel IB-IIA patients a priori, as we considered them to be ambiguous cases for evaluating the EZ implantation. The MNI cohort included the earliest and latest available nights satisfying the inclusion criteria (see

Supplementary Fig. S1), resulting in the selection of 90 overnight recordings from patients with drug-resistant focal epilepsy (54% females; mean age $32.3 \pm 10.8$). Among them, seventeen patients had a normal MRI. The CHUGA cohort consisted of 26 patients (53.9% females; mean age $32.2 \pm 15.7$) and segments from 26 nights, with eleven patients having a normal MRI. The main analysis utilized the latest recording post-implantation (obtained on average $10.3 \pm 4.6$ days after electrode insertion) based on the assumption that segments with higher IED rates, as commonly observed after seizures when lowering of antiseizure medication[21], are more accurate in predicting the EZ[22]. Initially conducted on the MNI cohort, the analysis was later validated on the CHUGA cohort. For intra-patient analysis, a one-hour segment from the beginning of the SEEG investigation was chosen (average $4.1 \pm 1.7$ days post-implantation). The patient demographics are shown in Supplementary Table S1.

### Constructing the spatial system for perturbation analysis

The spatial perturbation (SP) framework evaluates the response of a spatial system constructed from channel-level epileptic features to a perturbation of this system. The IED-$\gamma$ rate was used as our previous work showed that it localizes the EZ with high specificity[11]. The spatial system is constructed by coupling the IED-$\gamma$ rates with their distances to a spatial reference $\hat{\boldsymbol{\varphi}}_{sr}$ (Fig. 1a). In this study, we defined $\hat{\boldsymbol{\varphi}}_{sr}$ as the channel with the maximum IED-$\gamma$ rate (see "Methods"). The main premise for our framework is that we assume a *continuously decaying rate of the interictal biomarker with the distance to the region where this biomarker is maximum in case of a well implanted unifocal epilepsy*. Indeed, we see an example of this spatial coupling for a seizure-free patient and how this coupling could change if we simulate poor implantation of the SOZ using the same seizure-free patient (Fig. 1b). We empirically determined that the spatial system could be characterized by the goodness-of-fit of an exponentially decaying function, as shown in Fig. 2(a, b) for seizure-free and non-seizure-free patients, and as described in the "Methods". We then characterized the spatial system after applying a perturbation (Fig. 2c,d) and correlated the system's response with surgical outcomes. The complete methodology pipeline is depicted in Supplementary Fig. S2 and described in "Methods".

### Virtual removal of the SOZ disturbs the spatial system in seizure-free patients

A perturbation was applied to the spatial system by virtually removing the SOZ (Fig. 1c). The changes in the spatial system were then measured as described in the "Methods". To ensure dependency of the spatial system to the SOZ, $\hat{\boldsymbol{\varphi}}_{sr}$ was restricted to the SOZ before removal (BR) of the SOZ. We found that the virtual removal of the SOZ significantly reduces the spatial coupling in seizure-free patients in the MNI cohort (Wilcoxon's signed rank statistic ($W$) = 5; $p = 7.13 \times 10^{-4}$; Cliff's $d = 0.72$; $n = 17$; Fig. 3a). Similar results were found in the CHUGA cohort ($W = 20$; $p = 4.3 \times 10^{-3}$; $d = 0.46$; $n = 18$; Fig. 3a).

### Disturbance is inherent to the SOZ

We statistically tested whether the removal of the SOZ is indeed perturbing the system, and not just the reduction in the number of contacts. Therefore, we performed a bootstrapped removal of non-SOZ channels with 100 iterations, where the size of the randomly removed (RR) channels is the maximum proportion of the SOZ which could consistently be removed over all patients (as described in "Methods"). The power-law hypothesis is then tested and denoted as $\rho_{RR}$. We found no statistical difference between $\rho_{BR}$ and $\rho_{RR}$ in both centers (MNI: $W = 85$; $p = 0.69$; $d = 0.01$; Fig. 3a; CHUGA: $W = 111$; $p = 0.10$; $d = 0.02$; Fig. 3a).

### Disturbance is not present in non-seizure-free patients

We then investigated whether the virtual removal of the SOZ in the Engel IIB+ outcome patients will significantly affect the spatial

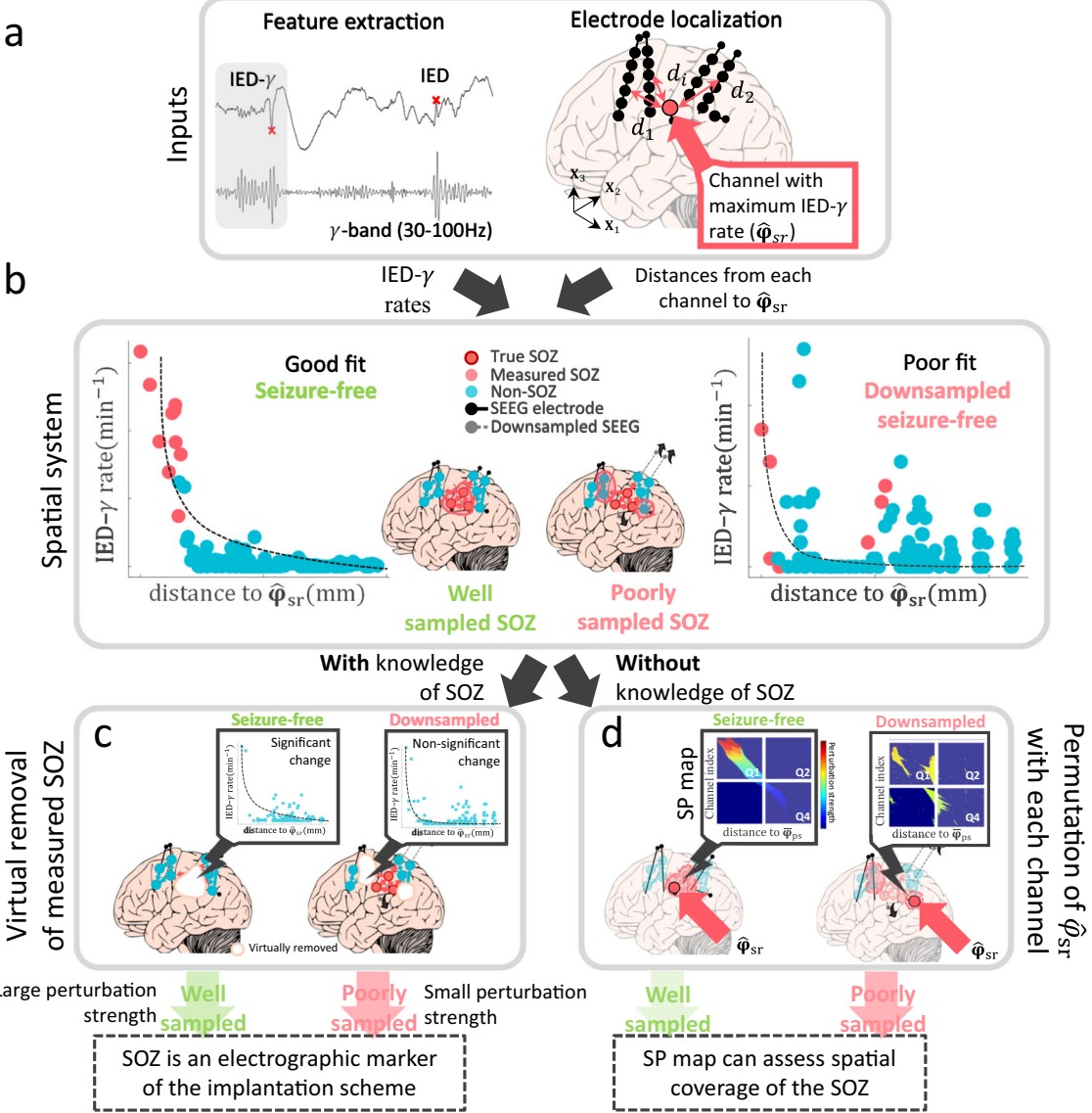

**Fig. 1 | The spatial perturbation framework. a** Inputs to the spatial perturbation framework. IED-γ are detected by finding IEDs with significant gamma activity preceding its onset as performed in refs. 11,20. SEEG channel coordinates are obtained by performing linear co-registration of post-implantation imaging with a template in normalized MNI space[45]. Distances are calculated between each channel and a spatial reference, defined as the channel with the maximum IED-γ rate. **b** The SP framework characterizes the implantation scheme by coupling IED-γ rates and their distance to the spatial reference. Examples are shown in a seizure-free patient, and the same patient, but three electrodes removed which are involved in the SOZ, to clearly simulate a poor implantation (i.e., downsampled seizure-free). **c** A perturbation was applied to the spatial system by virtually removing the measured SOZ. The change in the spatial coupling was quantified as the perturbation strength and was used to classify the implantation scheme. **d** Another kind of perturbation was applied to measure the implantation quality without knowledge of the SOZ by permuting the spatial reference and measuring the change in the

spatial coupling (i.e., perturbation strength). This is translated into the spatial perturbation map by ranking the channels by their perturbation strength (y-axis) and their distances to the centroid of the area with high perturbation strength (x-axis). The color intensity shown on the SP map is the perturbation strength (z-axis). *Note: The SEEG implantation shown in a is based on the implantation scheme of P2. Only a subset of six of 15 electrodes within the plane are shown, and the contacts are enlarged for clarity.* In (**b**), the seizure-free patient is used to demonstrate a well-sampled patient, and the same patient was used to demonstrate a poorly sampled SOZ by removing three electrodes which sampled the SOZ. This simulates an electrode configuration which fails to sample the 'true' SOZ. The spatial reference was restricted to the SOZ as done in the virtual-removal SP framework. All brain figures are based on real patient data, and are simplified for demonstration purposes. IED interictal epileptiform discharge, SOZ seizure-onset zone, SP spatial perturbation.

coupling as well. Given that these patients may have a mixed case of good and poor sampling (since a failed surgery may not necessarily be only due to poor sampling), we expect that the disturbance will not be as strong as that in Engel IA patients. As expected, the virtual removal of the SOZ did not significantly impact the spatial system in patients with Engel IIB+ outcome ($W = 228$; $p = 0.35$; $d = 0.18$; $n = 33$; Fig. 3b). Similar results were observed in the CHUGA cohort ($W = 24$; $p = 0.46$; $d = 0.13$; $n = 8$; Fig. 3b).

## Perturbation strength classifies the implantation scheme
Here we wished to use the system's response to the perturbation (i.e., virtual removal of the SOZ) to delineate the implantation quality of the SOZ. As described in the "Methods" section, the log of the absolute ratio between $\rho_{BR}$ and $\rho_{AR}$ was computed for each patient, defined as the perturbation strength. This measure attempted to describe the spatial sampling of the SOZ by classifying surgical outcomes. A threshold was computed using the MNI cohort and tested on the CHUGA cohort.

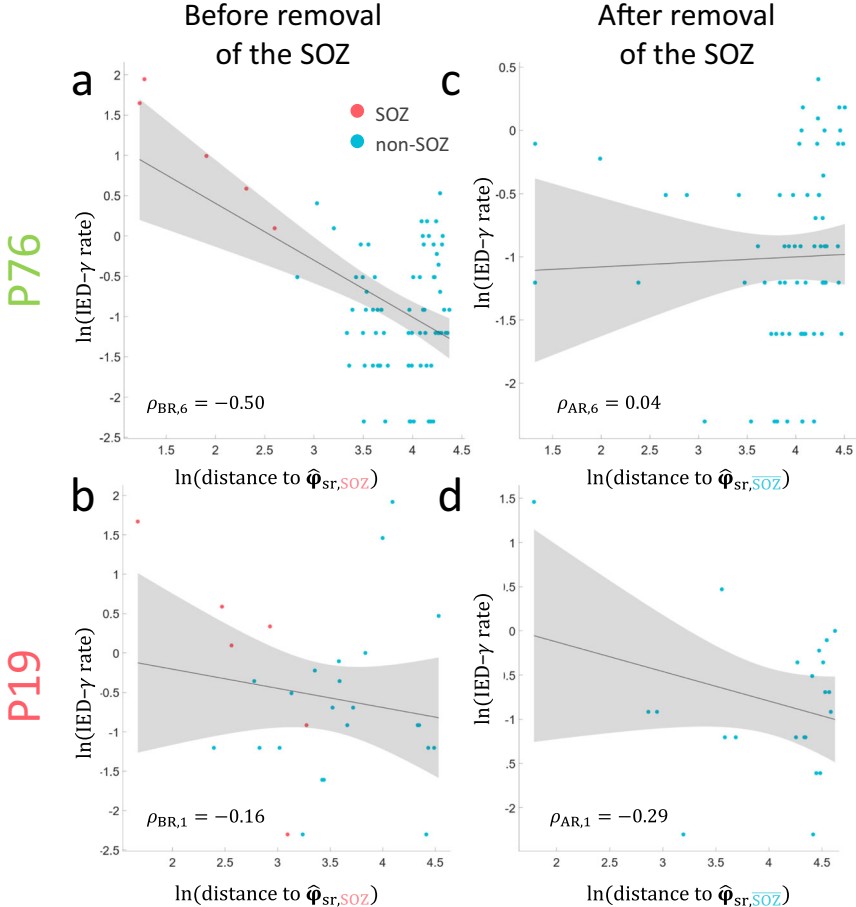

**Fig. 2 | Example of the virtual-removal SP framework applied on a seizure-free case.** Example of a scatter plot of a IED-γ rates computed from a continuous 10-minute interictal segment and distances between bipolar channel $\varphi_i$ and $\hat{\varphi}_{sr,SOZ}$ (i.e., channel with the maximum IED−γ rate within the SOZ) for patient P76 (seizure-free). The Pearson's correlation coefficient $\rho$ between the features and distances in log-log space, is computed (**a**) before removal with $\rho_{BR,6} = -0.50$ ($p = 5.4 \times 10^{-6}$) and (**c**) after removal with $\rho_{AR,6} = 0.04$ ($p = 0.77$). **b** The same spatial system computed for patient P19 (non-seizure-free) before removal of the SOZ with $\rho_{BR,1} = -0.16$ ($p = 0.41$) and (**d**) after removal with $\rho_{AR,1} = -0.29$ ($p = 0.19$). The shaded regions represent the 95% confidence interval of a linear fit. Source data are provided as a Source Data file. SOZ seizure–onset zone, IED interictal epileptiform discharge, BR before removal, AR after removal.

The perturbation strengths were compared between the two classes and were shown to be significantly different (shown in Fig. 4). In the MNI cohort, the perturbation strength was significantly higher in seizure-free patients ($n = 17$) than in non-seizure-free patients ($n = 33$) with a moderate effect (Wilcoxon rank sum statistic ($U$) = 573; $p = 4.4 \times 10^{-4}$; $d = 0.50$; area-under-the-curve (AUC) = 0.75). We saw the same results in the CHUGA cohort ($n = 18$ seizure-free, $n = 8$ non-seizure-free) ($U = 286$; $p = 0.02$; $d = 0.60$; AUC = 0.80). A threshold was selected based on the receiver operating characteristics (ROC) curve as described in the "Methods", obtaining a sensitivity of 0.76 and specificity of 0.61 using the MNI cohort resulting in $\tilde{\rho}^* \cong 1.20$. The optimized threshold $\tilde{\rho}^*$ was then tested on the CHUGA cohort for IED-γ prediction scores, resulting in a sensitivity of 0.61 and specificity of 0.75, respectively. There was no clear difference between MRI-positive and MRI-negative patients in either MNI or CHUGA cohorts.

**Perturbation strength is consistent throughout the recording duration**
To determine whether the perturbation strength may be impacted by the day of implantation, patients with full nights available at the beginning and end of the implantation (Engel IA: $n = 12$; Engel IIB+: $n = 28$) were analyzed. There was no statistical difference in the perturbation strength for IED-γ rates when an earlier day was considered for Engel IA patients ($W = 15$; $p = 0.064$; $d = 0.18$; $n = 12$) and Engel IIB+ ($W = 254$; $p = 0.25$; $d = 0.12$; $n = 28$), as shown in Supplementary

Fig. S3a. The perturbation strength was also not significantly correlated to the number of days between the two nights (Pearson's $\rho = -0.28$; $p = 0.38$; Supplementary Fig. S3b).

**SP framework produces an SP map**
The next step was to implement the SP framework without the need of recording the SOZ (Fig. 1d). Therefore, we applied a series of perturbations to the spatial system identified in the virtual-removal SP framework without constraining it to the SOZ (see "Methods" for details). We applied these perturbations by changing the spatial reference used when coupling IED-γ rates with their distances (see Fig. 5). This produced a series of perturbed spatial systems, where their responses can be measured and spatially ranked to construct an SP map.

After inspecting all SP maps of the good and poor outcome patients in the MNI cohort, we defined two main criteria for interpreting the SP map, which indicates a good implantation: (1) activation in the SP map exclusively near the diagonal of the map; and (2) a monotonically decreasing perturbation strength across the diagonal. A good implantation should result in a diagonal SP map, as it implies that the perturbation strength is focal, and that it decreases when moving away from the focal perturbation strength. However, a poor implantation would not result in a focal perturbation strength, and therefore, would not be a well-defined diagonal.

Figure 6a, b show clear examples of SP maps in two patients, P4 and P58, who had a good and poor implantation coverage,

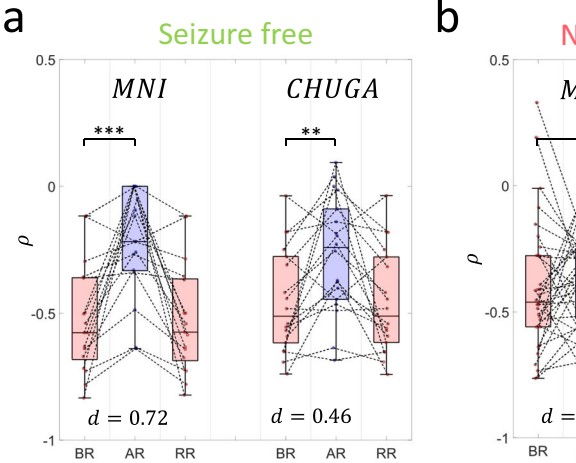

**Fig. 3 | Virtual removal of the SOZ in seizure-free and non-seizure-free patients.** The power–law hypothesis being tested on the spatial coupling of IED–γ rates before removal of the SOZ (BR), after removal of SOZ (AR), and random removal (RR) of non–SOZ contacts ($n = 20\%$ the size of the SOZ) for (**a**) Engel IA patients (MNI ($n = 17$): $\bar{\rho}_{BR}, \bar{\rho}_{AR}, \bar{\rho}_{RR} = -0.58\ (0.32)\ -0.22\ (0.33),\ -0.57\ (0.32)$; CHUGA ($n = 18$): $\bar{\rho}_{BR}, \bar{\rho}_{AR}, \bar{\rho}_{RR} = -0.51\ (0.34)\ -0.24\ (0.36),\ -0.51\ (0.34)$) and (**b**) Engel IIB+ patients (MNI ($n = 33$): $\bar{\rho}_{BR}, \bar{\rho}_{AR}, \bar{\rho}_{RR} = -0.46\ (0.28)\ -0.35\ (0.27),\ -0.45\ (0.28)$; CHUGA ($n = 8$): $\bar{\rho}_{BR}, \bar{\rho}_{AR}, \bar{\rho}_{RR} = -0.31\ (0.31)\ -0.34\ (0.17),\ -0.31\ (0.32)$). Virtually removing the measured SOZ significantly reduces the spatial coupling in Engel IA patients (MNI: $p = 7.13 \times 10^{-4}$, CHUGA: $p = 4.3 \times 10^{-3}$), with no difference in Engel IIB+ patients (MNI: $p = 0.35$, CHUGA: $p = 0.46$) using a two-sided paired Wilcoxon's signed rank test. There was no significant difference in the spatial coupling before virtually removing

the SOZ, and after randomly removing non-SOZ contacts in MNI ($p = 0.69$; two-sided paired Wilcoxon's signed rank test) and CHUGA ($p = 0.10$; two-sided paired Wilcoxon's signed-rank test) patients. Each point represents a patient after computing the median of correlations computed for each available 10-min segment. The center line of the boxplot represents the median, and the box limits represent the 25th and 75th percentile. The whiskers represent the complete range of values which are within 1.5 times the interquartile range. Summary statistics reported as median (IQR). Source data are provided as a Source Data file. BR before removal, AR after removal, RR random removal, IED interictal epileptiform discharge, IQR interquartile range, MNI Montreal Neurological Institute, CHUGA Grenoble Alpes University Hospital Center. Statistical significance is shown in asterisks: $^*p < 0.05$, $^{**}p < 0.01$, $^{***}p < 0.001$.

---

respectively. The $y$-axis of the SP map represents the bipolar channels, which are ranked by their perturbation strength in descending order. The $x$-axis represents the distances of each bipolar channel to the region of high perturbation strength. For patient P4, the implantation successfully identified a focal generator located in the left posterior cingulate gyrus. Surgical resection of the identified focus resulted in an Engel IA outcome. This can be seen in the map as a nearly diagonal matrix with decreasing intensity. In patient P58, the clinical chart indicated that the implantation revealed a generator in the left supplementary motor area. However, interictal abnormalities were very rare which may indicate that the implantation sampled the propagation zone. Surgical resection of the mid-portion of the left superior frontal gyrus resulted in an Engel IVC outcome. The SP map of P58 demonstrates a lack of a diagonal structure and no monotonical decrease in the intensity, therefore indicating that the SOZ may have been missed. Indeed, these patterns are consistent across patients from MNI and CHUGA as shown in Supplementary Figs. S4–7.

To prove the qualitative utility of the SP map, quantitative features were extracted. Three features were extracted (as described in "Methods"), using the first, second and fourth quadrant of the SP map, with the hypothesis that the clusters closest to the origin will model adequate sampling of the SOZ, and the cluster furthest away will model inadequate sampling of the SOZ. The optimal number of clusters identified by the k-means algorithm is two. The following are the resulting cluster centroids:

$$\hat{\mathbf{c}}_1 = (0.33, 0.11, 0.07)^T$$

$$\hat{\mathbf{c}}_2 = (0.44, 0.38, 0.25)^T$$

Indeed, $\hat{\mathbf{c}}_1$ and $\hat{\mathbf{c}}_2$ are significantly different from each other (see Supplementary Fig. S8). The scatter plots for MNI and CHUGA patient

cohorts are shown in Figs. 7a and 7b, respectively. The centroids were then used to classify surgical outcome, resulting in 64.7% sensitivity and 60.6% specificity in the MNI patient cohort. The centroids were tested on the CHUGA patient cohort, resulting in a sensitivity of 55.6% and specificity of 75.0%. Therefore, we hypothesized that the model could delineate good and poor implantations, with the $\hat{\mathbf{c}}_1$ cluster classifying good implantations, and the $\hat{\mathbf{c}}_2$ cluster classifying poor implantations.

### Good implantations in non-seizure-free patients are indicative of insufficient resections of the SOZ

We aimed to further investigate the results of our model in relation to variations in surgical outcome. Surgical failure can occur due to an incomplete resection, a more widespread SOZ than originally assumed, as well as poor sampling[10,23]. To control for these variations, an epileptologist assessed whether patients in the MNI cohort had incomplete resections of the measured SOZ (see "Methods"). Indeed, their implantation might have been successful; however the close proximity of the SOZ to functional cortex could have restricted the extent to which the measured SOZ could be removed, likely explaining the poor surgical outcome. As such, we will define resections which did not completely remove the measured SOZ *as incomplete resections*. To investigate this factor in the results, we checked how many of our patients were classified by the algorithm as well-sampled and a had poor outcome. We found that 62% (8/13) of these patients had incomplete resections of the measured SOZ.

Due to this proclivity of incomplete resections to be classified as well-sampled, we wanted to take a data-driven approach to mitigate this issue. To do so, we ensured that we only included patients who were unlikely to have an incomplete resection due to functional considerations. Thus, we only considered patients above the upper quartile of the resected SOZ volume distribution, extracted exclusively from MNI patients, who were marked as having an incomplete resection by a clinical expert who was blind to the results of this study. This resulted in a threshold of 3.305 cm³ of the resected SOZ volume (see

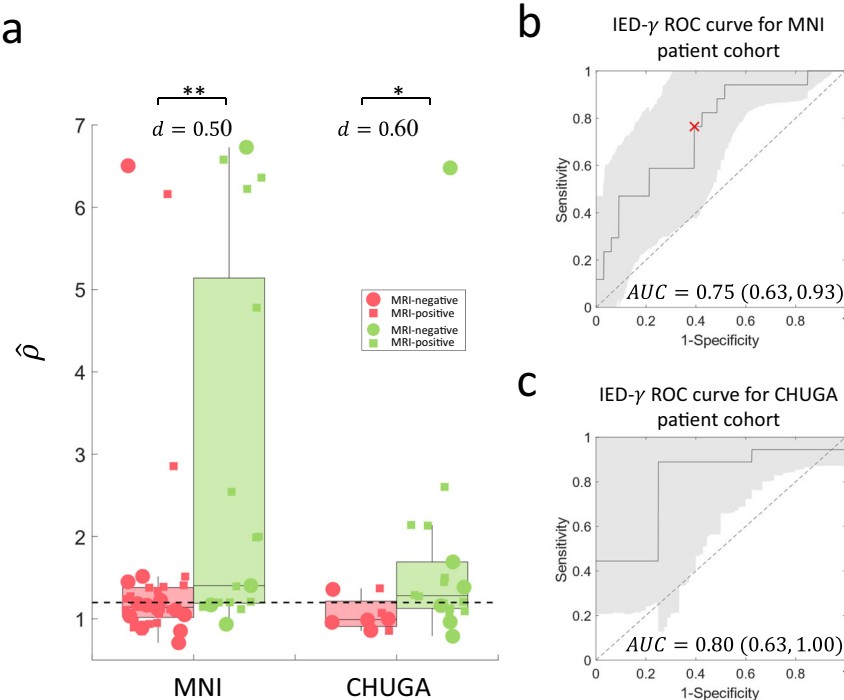

## Perturbation strengths

**Fig. 4 | Perturbation strength predicts surgical outcome.** The perturbation strength is computed for each patient (described in "Methods"). **a** In both centers, we see that seizure-free patients have a significantly higher perturbation strength (MNI ($n = 17$): $\hat{\rho} = 1.40$ (3.95); CHUGA ($n = 18$): $\hat{\rho} = 1.28$ (0.56)) compared to non-seizure-free patients (MNI ($n = 33$): $\hat{\rho} = 1.14$ (0.36); CHUGA ($n = 8$): $\hat{\rho} = 0.99$ (0.31)) with a difference of 0.26 ($p = 4.4 \times 10^{-4}$) and 0.29 ($p = 0.02$) using a two-sided Wilcoxon rank sum test for MNI and CHUGA patients, respectively. Non-seizure-free and seizure-free patients are shown as red and green dots, respectively. MRI-negative patients are represented as a circle, and MRI-positive are presented as a square. The center line of the boxplot represents the median, and the box limits represent the 25th and 75th percentile. The whiskers represent the complete range of values which are within 1.5 times the interquartile range. **b** The AUC for classifying the implantation scheme via the patients' surgical outcome is 0.75 (95% CI: 0.63,0.93). A threshold $\widetilde{\rho}^{*} \cong 1.20$ was selected in the MNI cohort to achieve a specificity of 0.61 and sensitivity of 0.76, as shown in the ROC curve. **c** The AUC on the CHUGA patient cohort is 0.80 (95% CI: 0.63,1.00). $\widetilde{\rho}^{*}$ is tested on the CHUGA cohort, obtaining a sensitivity of 0.61 and specificity of 0.75. The threshold is visualized as a horizontal dotted line in (**a**). Summary statistics are reported as median (IQR). Source data are provided as a Source Data file. AUC area under the curve, IED interictal epileptiform discharge, IQR interquartile range, MNI Montreal Neurological Institute, CHUGA Grenoble Alpes University Hospital Center, CI confidence interval, ROC receiver operating characteristics. Statistical significance is shown in asterisks: *$p < 0.05$, **$p < 0.01$.

"Methods"). After utilizing this threshold to exclude patients with incomplete resections of the SOZ, we recalculated the performance of our model, which resulted in a corrected sensitivity and specificity of 0.70 and 0.89 in the MNI cohort (Engel IA = 10, Engel IIB$^+$ = 9), and 0.75 and 1.00 for the independent test cohort (CHUGA) (Engel IA = 8, Engel IIB$^+$ = 4). In addition, we tested our model with different thresholds (70th, 80th and 85th percentile) and observed no changes in the model performance (Supplementary Table S2).

### Ranked SP framework models probability of adequate EZ implantation

A probability model was developed using the Euclidean distances from the good ($\hat{c}_1$) and poor ($\hat{c}_2$) implantation clusters (see "Methods"). In the MNI patient cohort (Fig. 7c), seizure-free patients ($n = 17$) are significantly more likely to have sampled the SOZ than non-seizure-free patients ($n = 33$) ($U = 552$; $p = 0.02$; $d = 0.42$; AUC = 0.71). Figure 7d shows that a similar classification performance was found in the CHUGA patient cohort (Engel IA = 18, Engel IIB$^+$ = 8) ($U = 271$; $p = 0.13$; $d = 0.39$; AUC = 0.69). After correcting for patients with insufficient resections, the probability that the SOZ was adequately sampled was significantly larger in seizure-free ($n = 10$) compared to non-seizure-free patients ($n = 9$) in the MNI cohort ($U = 137$; $p = 1.5 \times 10^{-3}$; $d = 0.82$; AUC = 0.91; Fig. 7e). A similar classification performance was found in the CHUGA cohort ($U = 62$; $p = 0.11$; $d = 0.63$; AUC = 0.81; Fig. 7f). We found no statistical significance in the CHUGA cohort given the low number of patients remaining after correcting for incomplete resections (Engel IA = 8, Engel IIB$^+$ = 4). We further tested the model with different thresholds (70th, 80th and 85th percentile) and observed no changes in the model performance (Supplementary Table S3).

### Poor implantations record larger SOZ volumes

Given the various reasons for poor surgical outcome, we found that MNI patients classified as poorly sampled tended to have a larger SOZ volume with moderate effect than those classified as well sampled ($U = 523$; $p = 0.09$; $d = 0.29$; $n = 26,24$; Fig. 8a). In addition, we compared seizure-free and non-seizure-free patients who were classified as having good implantations and found that the SOZ volumes were not different ($U = 136$; $p = 0.95$; $d = 0.02$; $n = 11,13$; Fig. 8b). The trend found in the SOZ volume between the two clusters motivated us to perform the same investigation after correcting for insufficient resections. Therefore, after correcting for insufficient resections, we found that patients classified as poorly sampled had a significantly larger SOZ compared to patients classified as well-sampled in the MNI cohort ($U = 53$; $p = 0.03$; $d = 0.61$; $n = 8,11$; Fig. 8c). We found the same trend in CHUGA patients ($U = 28$; $p = 0.09$; $d = 0.61$; $n = 6,6$; Fig. 8d). These results imply that in unifocal epilepsies, the SOZ volume measured by the SEEG is volumetrically bounded (first quartile = 3.3 cm$^3$, third quartile = 5.5 cm$^3$). However, when the SOZ is poorly sampled, the

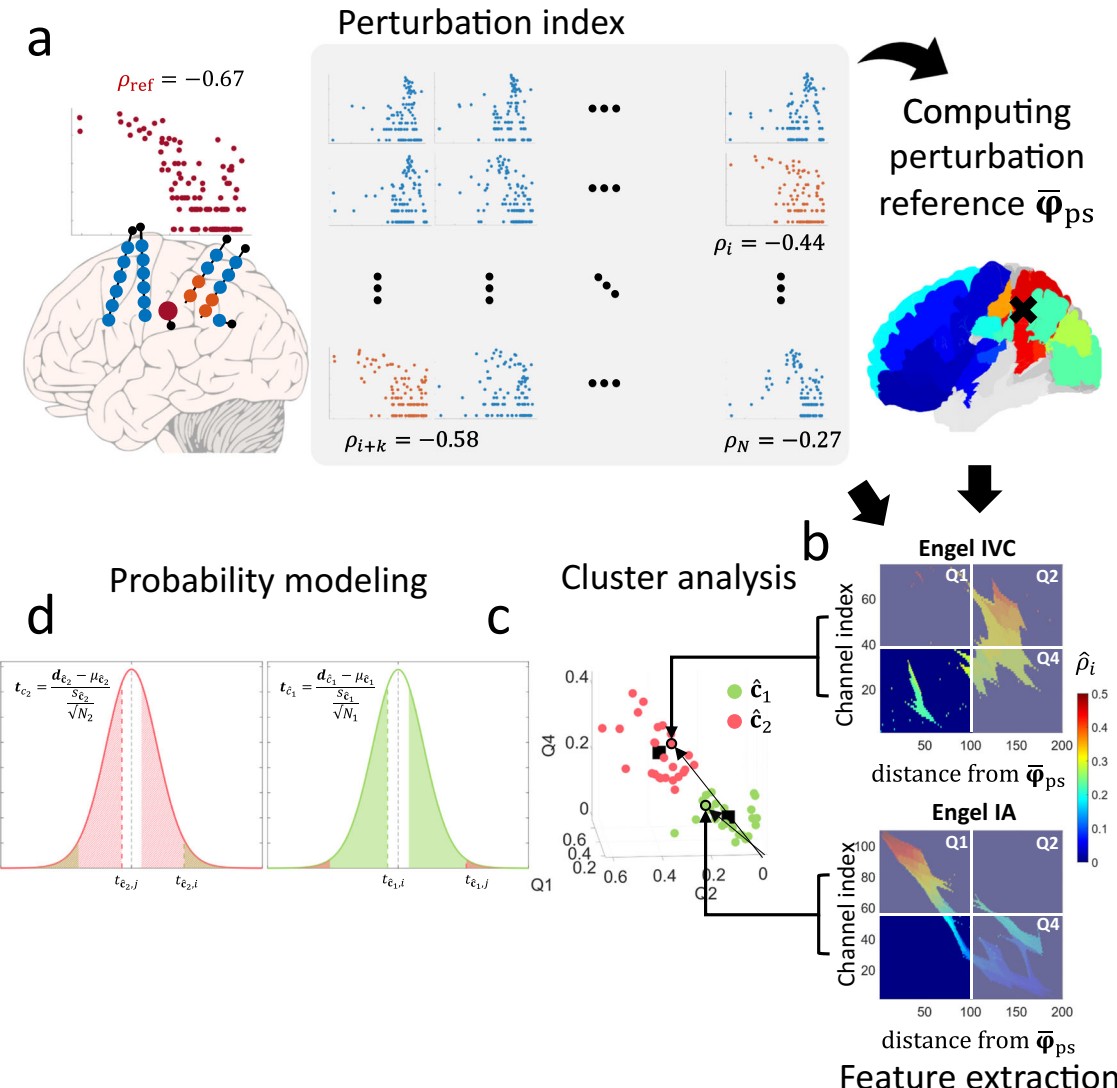

**Fig. 5 | Ranked SP framework pipeline.** The overall pipeline for constructing a model to determine if the SOZ was adequately sampled. **a** The perturbation strength is computed as the difference between $\rho_i$ and $\rho_{ref}$, where $\rho_{ref}$ is the Pearson's correlation obtained with the spatial system constructed using the channel with the highest IED-γ rate (without restriction to the SOZ) as its defined spatial reference. To measure the system's response to the series of perturbations on the spatial reference, it is excluded prior to computing $\rho_i$. $\rho_i$ is the Pearson's correlation between the IED-γ rate and distance from all channels to channel $c_i$ for $i = 1, 2, \ldots, N$ in log–log space, where $N$ denotes the total number of channels in the implantation scheme. The perturbation centroid is then computed as the mean of all channel coordinates with a perturbation strength greater than the $70^{th}$ percentile. This centroid will be considered as a proxy of the SOZ (Supplementary Fig. S9). **b** The Euclidean distance between the perturbation strength and the perturbation centroid is calculated and translated into a 2-D image with 200 bins. The rows of the image are sorted in ascending order, and image processing techniques are applied to spatially 'close' sparse regions that are in proximity. **c** Three features are extracted from the resulting image, which represents the mean of all positive perturbation indices in quadrants 1, 2, and 4 of the SP maps. Red dots denote patients classified as poor sampling, and green dots denote patients classified as well-sampled. **d** The clusters were used to develop a probability model that a patient is in cluster 1, but not in cluster 2. Classification results are shown as outlined points, and surgical outcome is the filled point (e.g., green outline and red fill is a non-seizure-free patient classified as well-sampled). IED interictal epileptiform discharge, SOZ seizure-onset zone, TP true-positive, FP false-positive, EZ epileptogenic zone.

SEEG may record the propagation network instead. This would result in marking a more widespread SOZ, which encompasses the propagation zone instead of the 'true' SOZ, resulting in a larger recorded SOZ volume.

**Seizure-freedom from poor implantations potentially linked to resections made in the unmeasured SOZ or the propagation zone**

Seizure freedom may not always be the result of a good implantation, since surgeons usually remove more tissue than only the SOZ. The model classified two sub-groups within the group of seizure-free patients: seizure-free patients classified as well-sampled, and seizure-free patients classified as poorly sampled. We found that the percentage of the SOZ resected was lower in seizure-free patients classified as having a poor sampling of the SOZ (Spearman's $\rho = -0.76; p = 4.5 \times 10^{-4}$; $n = 17$; Fig. 9a), while there was no difference in the resection volumes ($p = 0.33; n = 17$; Fig. 9b). These results potentially indicate that for seizure-free patients classified as poorly sampled, resections were either made in the SOZ not measured by the SEEG or in the propagation zone, either of which resulted in seizure freedom. Indeed, resections in the propagation zone may result in seizure-freedom. A previous study computationally demonstrated that optimized resections in the

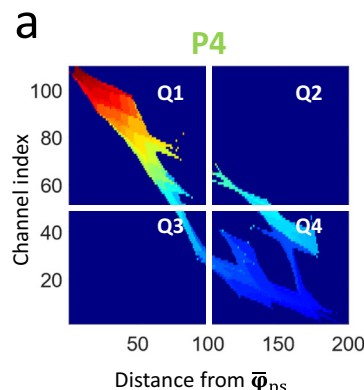
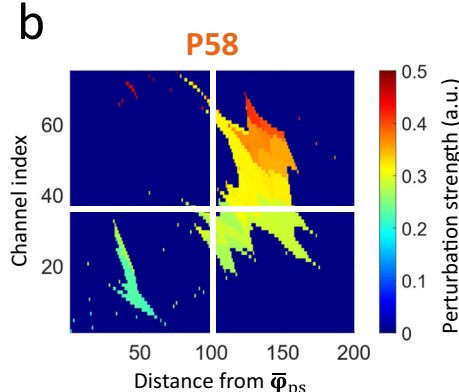

**Fig. 6 | Example of the SP maps.** Example images of the SP map computed on (**a**) patient P4 (seizure-free) and on (**b**) patient P58 (non-seizure-free). The mean of all positive pixels is computed on quadrants Q1, Q2, and Q4, forming a three-dimensional feature space. The hypothesis is that well-sampled patients should have a diagonal morphology with decreasing perturbation strength (shown in **a**), whereas poorly sampled patients have an ill-defined structure (shown in **b**). In a well-sampled patient, this would be represented by a small mean perturbation strength in Q2, and Q4, with a moderate mean perturbation strength in Q1. If the measured SOZ is ill-defined (as in P58), we would see activations in Q2 and Q4 which were not present in P4. SP spatial perturbation.

propagation zone can disrupt seizure propagation, while keeping the EZ intact[24]. We did not find these results in non-seizure-free patients: there were no significant correlations in either the percentage of SOZ removed ($p = 0.87$; $n = 33$; Fig. 9c) or the resection volume ($p = 0.36$; $n = 33$; Fig. 9d).

### Evaluating the SP map with patient case studies
We performed case-specific analyses of three patients to validate the clinical applicability of the SP map.

**Patient example P2.** P2 was a 26-year-old female patient who underwent a SEEG investigation in 2018. MRI results indicated the presence of a lesion in the deep sulcus located adjacent to the posterior cingulate gyrus, equivocal of focal cortical dysplasia (FCD). Phase 1 investigation lateralized the focus to the right hemisphere, and localization of the generator towards either the right hemispheric mesio-frontal or mesioparietal structures. After insertion of the SEEG electrodes, the recorded SOZ confirmed that the lesion suspicious for FCD was the seizure generator. The patient underwent a right mesio-parietal resection resulting in an Engel IA outcome. Figure 10a illustrates a very clear case of a well implanted SOZ. The SP map closely resembles an ideal diagonal matrix. This is represented by the patient's proximity to $\hat{c}_1$ in the feature space. Consequently, the model indicates that the SOZ implantation was of excellent quality. This assessment aligns with the successful sampling of the FCD during implantation, confirming its role as the seizure source.

**Patient example P49.** P49 was a 40-year-old male patient who underwent a SEEG investigation in 2011. The MRI indicated atrophy and gliosis in the bilateral parietal-occipital cortex. Phase 1 investigation was inconclusive for the lateralization of the focus, with either a parietal or temporo-parietal focus. Therefore, the patient underwent bilateral implantation of the temporo-parietal regions. Multiple seizures with a single type of semiology were captured that were either EEG seizures, pure clinical seizures, or electro-clinical seizures with clinical manifestations preceding the first changes in the ictal SEEG, and localized mainly to the right temporal region. The patient underwent a palliative right temporal resection resulting in an Engel III outcome. Figure 10b illustrates the patient's map. Qualitatively, the diagonal matrix is not well defined, and does not decrease in intensity. In the feature space, we see that it is near $\hat{c}_2$ and significantly far from $\hat{c}_1$, indicating that the EZ was not sufficiently sampled. The model suggests that the EZ was partially sampled (probability of 49%). Given the electro-clinical data, the seizure with clinical manifestations preceding ictal changes suggests that the implantation is recording propagation activity. Additionally, the absence of EEG changes (i.e., pure clinical seizures) is also indicative of a missed focus and can be explained as the implantation had only sampled one of many propagation pathways. This is evident on the map as islands with significant perturbation indices without spatial decay, indicating a lack of a measured epileptogenic structure, consistent with the electro-clinical findings.

**Patient example P80.** P80 was a female patient aged 21 who underwent SEEG investigations in 2014 and 2021. The first implantation failed to identify a focal generator, and no surgery was attempted. The second implantation, however, revealed a focal generator in the right posterior temporo-occipital cortex. The patient underwent a surgical resection of the right lateral occipito-temporal cortex, resulting in an Engel IB outcome.

The one-hour segments were both extracted five days after each implantation. The SP map classified the first implantation as completely missing the epileptic focus, as shown in the feature space in cluster 2, being very far from $\hat{c}_1$. Regarding the second implantation, we see in the feature space, that the patient moved 42% closer to $\hat{c}_1$ (as shown in Fig. 10c, d). Although we see a significant improvement in the feature space after the second implantation, the model still classifies it as a sub-optimally sampled EZ, which aligns with the clinical impression. This may explain the patient's lack of complete seizure freedom (Engel IB). Posterior onsets typically have rapid propagation and are more widespread, which can make the precise localization of the focal generator challenging. The epileptologist involved in the SEEG placement has noted that the implantation nearly sampled the epileptogenic tissue but may have missed parts of the basal sulcus of the FCD.

## Discussion
In this study, we developed a framework to assess adequate SEEG sampling of the SOZ. The SP framework uses channel-specific epileptogenic features and transforms them into a spatial system without the need of recording seizures. We hypothesized that there is a continuously decaying rate of the interictal biomarker with the distance to the region where this biomarker is maximum in the case of a well sampled unifocal epilepsy. We also hypothesized that the change in the decay would be much larger in seizure-free patients, than in non-seizure-free patients. Indeed, the response of this spatial system to a perturbation was used to estimate the SOZ coverage. The three main findings of this study are the following: (1) the SEEG-identified SOZ encodes valuable information on the validity of the implantation; (2)

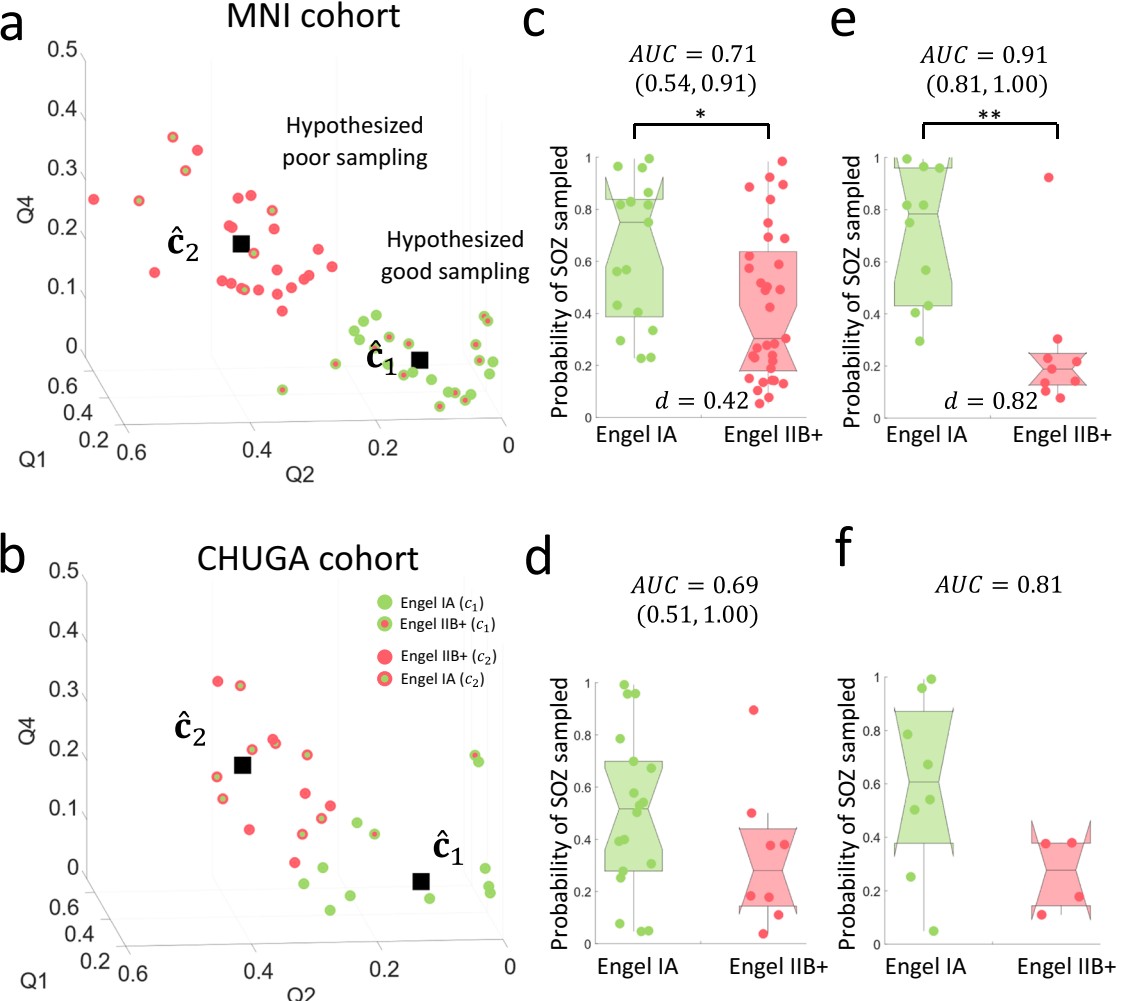

**Fig. 7 | Cluster analysis and probability of adequate SOZ implantation.** The 3-D feature space with the identified hypothesized clusters of good ($\hat{\mathbf{c}}_1$) and bad ($\hat{\mathbf{c}}_2$) sampling which were trained using (**a**) MNI cohort and validated in (**b**) CHUGA cohort. **c** The results of the probability model in the MNI patient cohorts. The probability of the Engel IA patients ($n = 17$) having a successful implantation is 0.75 (0.45) (median (IQR)), and is significantly larger ($p = 0.02$; two-sided Wilcoxon rank sum test) than Engel IIB+ patients ($n = 33$), which have a median probability of 0.30 (0.46). **d** Similar classification performance is found in the CHUGA cohort ($p = 0.13$; two-sided Wilcoxon rank sum test; Engel IA ($n = 18$): 0.52 (0.28); Engel IIB+ ($n = 8$): 0.28 (0.30)). **e** A clinical expert marked whether a patient in the MNI cohort had an incomplete resection due to functional considerations. A threshold was obtained (3.305 cm³) using the 75th percentile of the resected SOZ volume in patients with incomplete resections (marked by clinical expert) with the premise that lower volumes of the SOZ removed may indicate incomplete resections. After removing patients who did not have a sufficient resection of the SOZ, the AUC is corrected to 0.91 with 0.78 (0.53) and 0.19 (0.12) probability of adequate sampling

in Engel IA ($n = 10$) and Engel IIB+ patients ($n = 9$), respectively, and is significantly higher in Engel IA patients ($p = 1.5 \times 10^{-3}$; two-sided Wilcoxon rank sum test) **f** The threshold was validated on the CHUGA cohort, correcting the AUC to 0.81 with 0.61 (0.49) and 0.28 (0.23) probability of adequate sampling in Engel IA ($n = 8$) and Engel IIB+ patients ($n = 4$), respectively. However, this is not statistically significant ($p = 0.11$, two-sided Wilcoxon rank sum test), possibly due to the small number of patients remaining after correction. The center line of the boxplot represents the median, and the box limits represent the 25th and 75th percentile. The whiskers represent the complete range of values that are within 1.5 times the interquartile range. The notch around the center line of the boxpot represents the 95% confidence interval of the median. Source data are provided as a Source Data file. MNI Montreal Neurological Institute, CHUGA Grenoble Alpes University Hospital Center, AUC area under the curve, TP true-positive, FP false-positive, FN false-negative, TN true-negative, Q quadrant, EZ epileptogenic zone, SOZ seizure-onset zone. Statistical significance shown in asterisks: *$p < 0.05$, **$p < 0.01$, ***$p < 0.001$.

the SP framework classifies adequate SOZ coverage using the response of the spatial system to a perturbation; (3) the creation of a SP map facilitates visual interpretation of the implantation scheme.

This study proposes an SP framework which assesses the SEEG implantation of the SOZ in two ways: (1) virtual removal of the SOZ; (2) permutation of the spatial reference, defined as the channel with the maximum IED-$\gamma$ rate, with each channel in the SEEG implantation scheme. The first method (we call it virtual-removal SP framework) tests the power-law hypothesis before and after SOZ removal. We found that seizure-free patients had a stronger perturbation strength after virtually removing the SOZ when compared with non-seizure-free patients and could classify implantation adequacy with an AUC of 0.75

and 0.80 for MNI and CHUGA patient cohorts. The results from the virtual-removal SP framework therefore indicate that the SOZ measured by SEEG encodes valuable information on the implantation scheme. The second method (the ranked SP framework) attempts to include this information and applies the framework without the need to record seizures. This was done by applying a series of perturbations by permuting the spatial reference with all the channels available in the implantation. The system's response to these series of perturbations was measured, and spatially ranked to construct the SP map of the SEEG implantation. The spatial ranking process can be seen as a perturbation on a meta-system, which is the combination of all spatial systems perturbed using each channel of the SEEG. We inherently

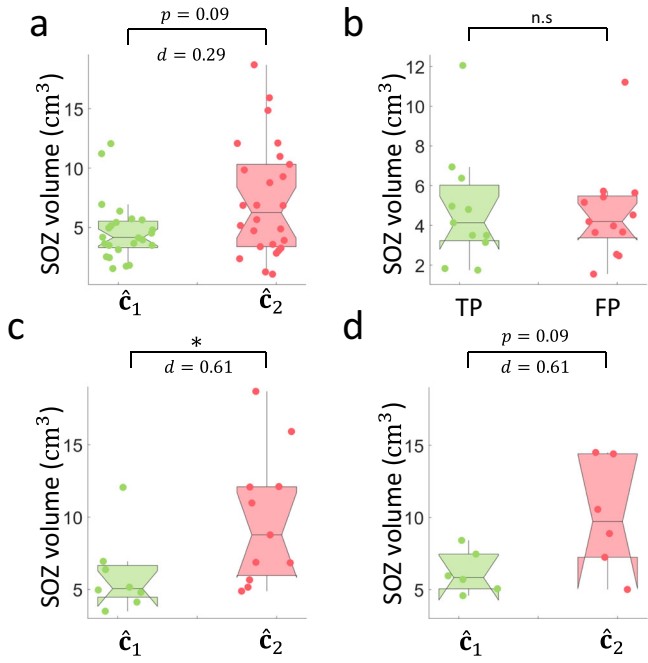

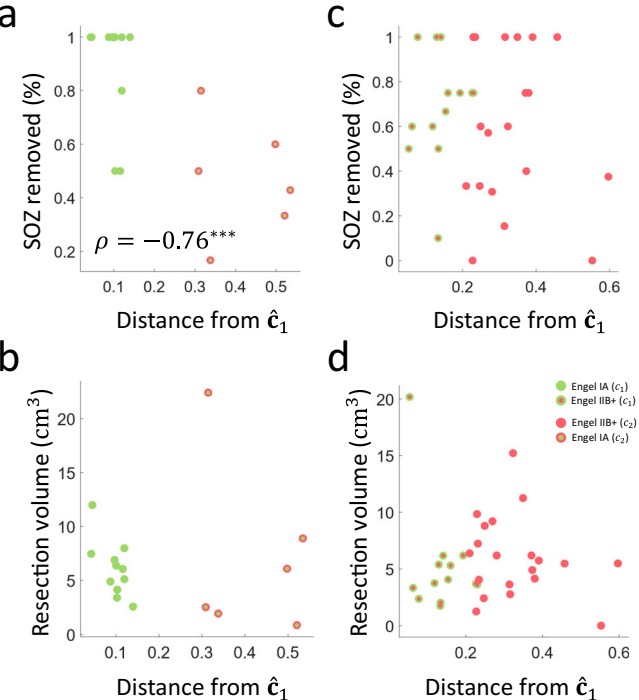

**Fig. 8 | Poor implantations record larger SOZ.** The SOZ volumes were estimated as described in "Methods". **a** The SOZ volume tends to be larger ($p = 0.09$; two-sided Wilcoxon rank sum test) when comparing the patients in cluster 2 (poorly-sampled; $n = 26$; median volume $= 6.3(6.9)$cm$^3$) with the patients in cluster 1 (well-sampled; $n = 24$; median volume $= 4.2(2.2)$cm$^3$), although not statistically significant. **b** The SOZ volumes are not significantly different ($p = 0.95$; two-sided Wilcoxon rank sum test) for seizure-free ($n = 11$; median volume $= 4.1(2.8)$cm$^3$) and non-seizure-free patients ($n = 13$; median volume $= 4.2(2.1)$cm$^3$) classified as well sampled. **c** After correcting for insufficient resections (i.e., removing patients with resected SOZ volume below 3.3 cm$^3$), the SOZ volume is significantly larger ($p = 0.03$; two-sided Wilcoxon rank sum test) in MNI patients classified as poorly sampled ($n = 11$) (median volume $= 8.8(6.1)$cm$^3$) than those who are classified as well-sampled patients ($n = 8$; median volume $= 5.1(2.2)$cm$^3$), with the same trend in (**d**) CHUGA cohort ($p = 0.09$; two-sided Wilcoxon rank sum test; $\hat{c}_2(n = 6) : 9.7(7.2)$cm$^3$; $\hat{c}_1(n = 6) : 5.8(2.4)$cm$^3$). However, this is not significant due to low statistical power. The center line of the boxplot represents the median, and the box limits represent the 25$^{\text{th}}$ and 75$^{\text{th}}$ percentile. The whiskers represent the complete range of values that are within 1.5 times the interquartile range. The notch around the median center line of the boxpot represents the 95% confidence interval of the median. The summary statistics are reported as median (IQR). Source data are provided as a Source Data file. IQR interquartile range, TP true positive (well-sampled seizure-free patients), FP false positive (well-sampled non-seizure-free patients), TN true negative (poorly sampled non-seizure-free patients), SOZ seizure-onset zone. Statistical significance is shown in asterisks: *$p < 0.05$.

**Fig. 9 | Seizure-freedom from poor implantations linked to resections made in the unmeasured seizure-onset zone or propagation zone. a** The percentage of SOZ removed as a function of distance from $\hat{c}_1$ for seizure-free (green-fill) patients in $\hat{c}_1$ (in green-outline) and $\hat{c}_2$ (red-outline). There is a strong inverse correlation for seizure-free patients (Spearman's $\rho = -0.76$; $p = 4.5 \times 10^{-4}$; permutation test). **b** No statistically significant correlation was found for non-seizure-free patients (red-fill) in $\hat{c}_1$ (green-outline) and $\hat{c}_2$ (red-outline). **c** Similarly, there was no statistically significant correlation in resected volume for seizure-free patients and **d** non-seizure-free patients as a function of distance from $\hat{c}_1$. This implies that there is no difference in the amount of tissue resected for these patients indicating that the propagation zone or the unmeasured SOZ may have been resected, enabling seizure-freedom despite the poor implantation of the SOZ. In one patient, SEEG did not allow to identify the generator. Based on all available clinical information he underwent palliative surgery, explaining why none of the apparently SOZ channels has been removed during surgery. Source data are provided as a Source Data file. SOZ seizure-onset zone. Statistical significance shown in asterisks: ***$p < 0.001$.

perturbed this meta-system using a proxy of the SOZ (region with high perturbation strength), as we found that the perturbation strength is significantly higher in the SOZ than non-SOZ (see Supplementary Fig. S9).

To strengthen the foundation of our framework on the use of IED-$\gamma$ activity, we investigated the virtual-removal SP framework using more traditional interictal biomarkers such as IEDs and ripples (80–250 Hz) (see Supplementary Fig. S10). In theory, the framework can be applied to any interictal marker with the requirement that it is specific to the EZ. We found that IEDs constructed a sub-optimal spatial system given that they are rather unspecific to the EZ[15]. Virtually removing the SOZ perturbs the IED spatial system less than in the IED-$\gamma$ spatial system in both, MNI and CHUGA patient cohorts. We also found that ripples perform poorly in the virtual-removal SP framework. The reason for the poor perturbation profile of the ripple-rate spatial system could be due to the confound of physiological ripples, given large spatial variability across different brain regions[25,26].

In addition, we found that the results were not correlated with the presence of a lesion (i.e., MRI-positive) when correlating the perturbation strength with surgical outcome in the virtual-removal SP framework (see Fig. 4). Although MRI-positive cases should have a higher likelihood of a good implantation, it has been shown that MRI lesions do not necessarily correlate with surgical outcome in temporal lobe epilepsy evaluated by bitemporal intracranial implantations[25]. Indeed, while most of the implanted patients at the MNI underwent advanced MRI imaging analysis, previous findings from our group showed no difference in outcome in case of an identified MRI lesion[27].

The SP map can help clinicians to qualitatively assess the implantation scheme by: (i) checking proximity of the map to an ideal diagonal matrix; and (ii) checking for a decreasing perturbation strength across the diagonal. We were able to develop these criteria, as the SP map essentially tests three things: (1) whether the maximum IED-$\gamma$ channel satisfies the power-law hypothesis; (2) whether the perturbation strength delineates a focal region; and (3) whether the maximum IED-$\gamma$ channel is within this focal region. By spatially ranking the perturbation strength, it allows one to evaluate the diagonality of the matrix and interpret the success of the implantation. An example of the potential use of evaluating the implantation quality of the SOZ is patient P80, who had two implantations. The SP map reveals the

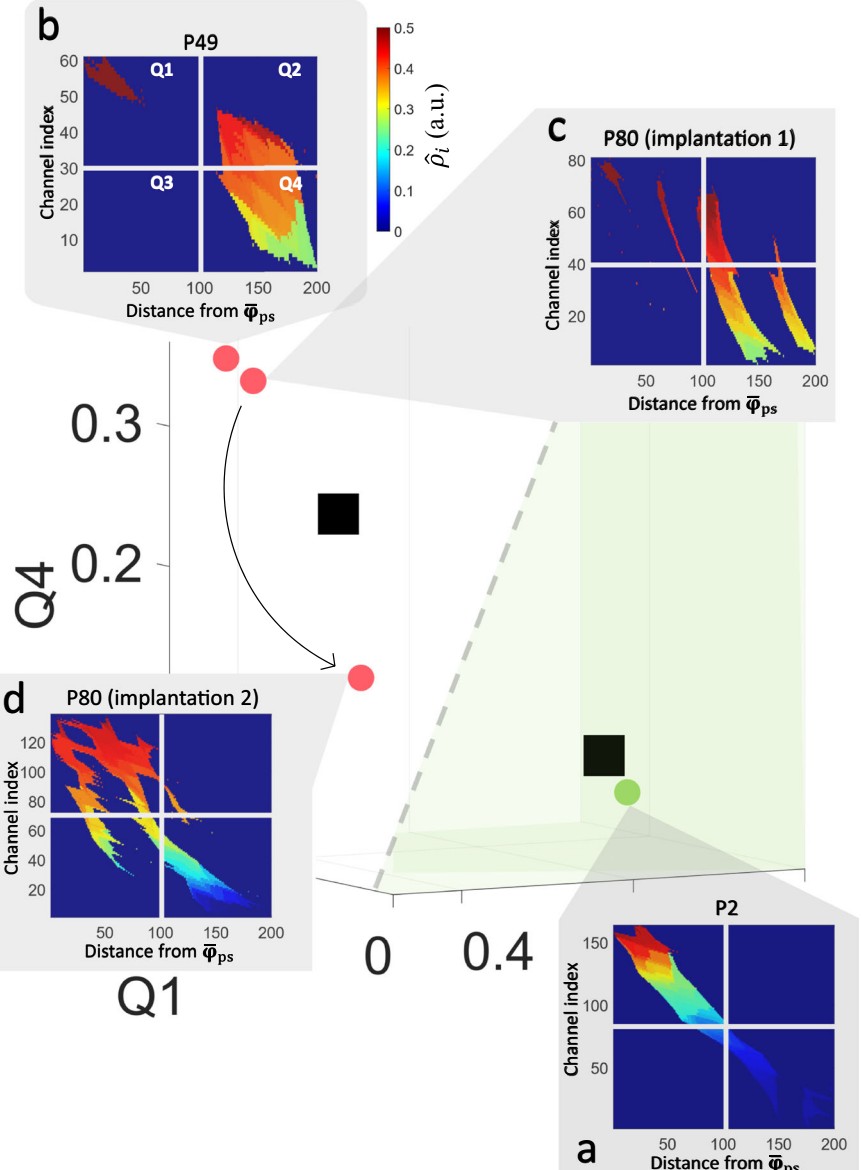

**Fig. 10 | Scatter plot of case study patients in the SP map feature space. a** Patient P2 had a lesion which was implanted by the SEEG and confirmed to be the seizure generator, resulting in seizure-freedom after surgery. **b** Patient P49 had clinical manifestations which preceded initial ictal EEG changes, suggesting a missed EZ implantation. **c** Patient P80 underwent SEEG implantation in 2014, and no surgery was performed due to a lack of an identifiable generator. **d** A second implantation was performed for P80 in 2021, causing the patient to move towards the decision boundary in the feature space. Experts noted that the implantation enabled the identification of a temporo-occipital generator, however, it may have missed the bottom sulcus of the FCD. This explains why the patient is only near the decision boundary (gray dotted line). The green-shaded region represents the good sampling cluster region. Source data are provided as a Source Data file.

progression from a very poor implantation to a partial coverage of the SOZ. Another example of the clinical utility of the map is the case of a missed generator in patient P49, evident by clinical manifestations preceding the first ictal changes. The map highlighted that the bilateral implantation failed to identify a clear focus, and therefore, is represented as two islands with large perturbation indices that do not decay and are not spatially coherent.

This study directly tackles the issue of sub-optimal spatial sampling of the EZ using simple measures derived from SEEG recordings. This was done by evaluating the implantation, and its ability to record an SOZ which fits certain priors on focality and presence of interictal markers of epileptogenicity. Previous studies have only attempted to refine the localization of the epileptic focus[13,28], or aimed to find spiking regions or SOZ missed by the SEEG[14,17–19]. Although finding novel spiking regions is a valuable tool to study the interictal network of spikes, they are also known to

delineate the irritative zone, which encompasses a larger region than the EZ[15]. In addition, while delineating seizure onset regions missed by the SEEG can play an important role in determining whether the electrode configuration has sampled the 'true' SOZ, this alone does not provide prospective value of whether the implantation was sufficient for delineating and surgically removing the measured SOZ to achieve seizure-freedom.

One of the key elements of this present study is the use of perturbations on a spatial system constructed from interictal biomarkers, allowing us to classify the implantation without knowledge of surgical information, outcomes and without the need to record seizures. A similar approach has been utilized for surgical outcome prognostication, which uses the coupling strength between structural connectivity of SEEG contacts as informed by diffusion-weighted MRI and their functional connectivity[29]. They have shown that 'virtually resected' channels which increase the structural-functional coupling overlap

more with the resection volume in seizure-free patients than in non-seizure-free patients.

The visualization offered by the SP map may allow one to interpret it as a 'fingerprint' of the implantation scheme. Current literature lacks a fingerprint that visualizes the quality of the implantation of the SOZ. Previous fingerprints were developed and validated for the localization of the EZ. They essentially visualize the 'epileptogenicity' of a given channel by their spectro-temporal dynamics[30–32], or whether the dynamical system estimated from the given channel is 'fragile' (i.e., near a limit cycle)[33].

In this study, simple features were computed to quantify the utility of our 'fingerprint'. As opposed to the supervised approach done in the literature[32–35], the main concept behind the feature extraction process in this study was to identify clear clusters in an unsupervised manner to classify the spatial sampling without the need of a ground truth. To validate these clusters, classification performance is later reported using surgical outcomes. As expected, the poor classification performance is the result of non-seizure-free patients who were well sampled but had insufficient resections due to functional considerations. In addition, we found that seizure-free patients had an inverse correlation between their distance from $\hat{c}_1$ in the feature space and the percentage SOZ removed, whereas no correlation was found when comparing with the resection volume. These results indicate that the seizure-free patients who were classified as having poor implantation may have had resections made in the propagation zone or in the unmeasured SOZ (since the resected cavity is typically larger than the presumed EZ).

A key limitation of this study is the lack of a clear ground truth, a general problem found in EZ localization research, as it is inherently assumed that the implantation was successful. We mitigated this issue by employing unsupervised techniques to develop a large sample-sized probability model, which was validated after correcting for patients with incomplete resections as marked by a clinical expert blind to the results of this study. Another potential limitation is the limited number of resected and/or SOZ channels, which may compromise the volume estimations. Notwithstanding, in the group of well-sampled patients, we found that seizure-free patients had a larger resected SOZ volume than non-seizure-free patients, with no significant difference in the SOZ volume. These findings demonstrate a good level of consistency despite the inaccuracies in volume estimation.

## Methods

### Patient and data selection

We screened a total of 72 consecutive focal drug-resistant epilepsy patients from the MNI and 37 consecutive patients from the CHUGA who underwent SEEG sampled at 512 Hz or higher with subsequent open resective surgery between 2009 and 2019; all patients had a postsurgical follow-up duration of at least 1 year. The inclusion criteria required patients to have at least 10 min of continuous interictal non-rapid eye movement (NREM) sleep available for analysis[36], such that they are at least two hours away from clinical seizures and at least 5 min away from pure electrographic seizures. For robust training data, a maximum of 1 h of interictal data was selected from the MNI patient cohort, as it was shown that IED rates have spatiotemporal fluctuations over time[27].

Out of the 72 patients from the MNI and 37 from CHUGA, 50 and 26 patients were included into our analysis as per the selection criteria (see flowchart in Supplementary Fig. S1, and patient demographics in Supplementary Table S1). We excluded Engel IB-IIA patients a priori, as we considered them ambiguous cases considering EZ implantation evaluation. This study was approved by the respective research ethics boards (MNI REB IRB00010120, Cogepistim MR004 11.05.21 DRCI CHUGA). Written informed consent was obtained from all patients. Sex- and gender-based analyses was not included in the study design,

as they have not been shown to impact the outcome of epilepsy surgery[37–39].

The latest available overnight SEEG sampled at 512 Hz or higher was selected, as our most recent study showed that timing matters for interictal biomarker analysis, with segments with high IED activity being associated with higher accuracy for correct delineation of the EZ; usually IEDs are higher when seizures have been recorded[22]. For intra-patient evaluation, this night was contrasted with the earliest available overnight SEEG sampled at 512 Hz or higher. Since some contacts were removed later in the investigation, the intersection of the channels in both nights were considered for the analysis. Ten of the 50 patients were removed due to the unavailability of two nights fulfilling the selection criteria. The flowchart for these selected patients is also shown in Supplementary Fig. S1.

The MNI recordings were obtained using either Harmonie (Montreal, Quebec, Canada) (Stellate, with a low-pass filter set at 500 Hz) or Nihon-Kohden (Tokyo, Japan) EEG amplifiers (with a low-pass filter set at 600 Hz) and with either homemade MNI or commercial DIXI Medical (Besançon, France) electrodes. The CHUGA recordings were acquired using Micromed (Mogliano Veneto, Italy) EEG amplifiers, with either DIXI Medical or ALCIS (Besançon, France) electrodes, and with low-pass filters set at 200, 276, and 552 Hz for sampling frequencies of 512, 1024, and 2048 Hz[11].

### Definition of the 'well-sampled' class and SOZ

A board-certified epileptologist identified the SOZ for all patients, defined as the first unambiguous changes in the EEG at seizure onset independent of the fast activity content and seizure-onset pattern[27,40]. We defined a well-sampled patient as Engel IA patients, as they had an electrode configuration sufficient to identify and resect the presumed EZ, leading to their seizure freedom. While the positive class, concerning the adequacy of SOZ implantation, is well-defined, the negative class may suffer from lack of clarity due to other factors which may result in a poor surgical outcome such as incomplete resection of the epileptic tissue due to (i) its proximity to eloquent cortex, and (ii) palliative surgical approaches. However, despite these confounding variables within the non-seizure-free cohort, all patients with IIB+ outcome constitute the negative class to thoroughly test the methodology on a large sample size and to avoid introducing bias with respect to the reason for surgical failure.

### Selection of the interictal segments

Sleep scoring was performed fully automatically using the SleepSEEG algorithm (https://doi.org/10.5281/zenodo.7410501). SleepSEEG is an automatic sleep-scoring algorithm[41] which was shown to be able to score sleep data with a median of 78% agreement between those marked by two human experts, where N2 and N3 stages of sleep performed the best (sensitivity: 0.85, 0.87; specificity: 0.76, 0.94). We used the model, which is blind to epileptic activity, and applied it on around eight hours of overnight data to mark the sleep stages (wake (W); rapid-eye movement (R); N1; N2; N3) in 30-s epochs. Previous studies have demonstrated that IED rates are higher in non-REM sleep compared to wake and REM[42,43]. It was also shown that the interictal SEEG during NREM sleep can best localize the EZ[44]. Therefore, a minimum of ten continuous minutes of NREM data, that included the least number of W and R epochs, were selected. An example output of the algorithm is shown in Supplementary Fig. S2a.

### Pre-processing and feature extraction

All signals were analyzed using the bipolar montage. SEEG electrode coordinates were localized by co-registration of the post-implantation MRI to a template in normalized MNI space for group-level analysis[45]. A bipolar channel coordinate was computed as the midpoint between the two contact coordinates. Extra-cerebral and white matter channels were then identified

using the MICCAI atlas[46] described in a previous study[45,47] and removed from the analysis. In addition, channels with significant amounts of artifacts were visually identified and removed. The power-line noise was removed using a 60 Hz and 50 Hz notch filter for the MNI and CHUGA datasets. The IED-$\gamma$ rate feature is subsequently computed by detecting significant gamma activity (30–100 Hz), with at least three cycles preceding the onset of an IED (see refs. 11,20), as described in Supplementary Method S1. The code used to detect IED-$\gamma$ can be found in our GitHub repository (https://github.com/Lab-Frauscher/Spike-Gamma). The overall pre-processing and feature extraction pipeline is depicted in Supplementary Fig. S2.

## The SP framework

The SP framework involves the use of interictal biomarkers to evaluate the implantation of the SOZ. This is done by transforming these channel-level biomarkers into a spatial system. A perturbation is applied to this spatial system, and the response to this perturbation is measured and used to evaluate the implantation scheme. The interictal biomarker used in this study is the IED-$\gamma$ rate, as it demonstrated significant specificity to the EZ in our previous study[11]. The overview of the framework is illustrated in Fig. 1. In this study, a 'spatial system' refers to the coupling of interictal biomarkers with their spatial distribution. A 'perturbation' refers to the act of changing this spatial system. A 'response' refers to the measured change in the spatial system when it undergoes some perturbation.

**Development of the spatial system.** While it is true that the absence of channels with high IED rates would imply that the electrode configuration did not sample the epileptic source[9,27], it is difficult to use event rates only to measure 'sufficient' sampling of the epileptogenic tissue, as there will always be a channel with the highest rate. This motivated us to consider a spatial relationship, instead of considering the feature values alone. Patients are usually implanted to identify a single seizure focus. When investigating the distance from the epileptic tissue to each channel in the SEEG implantation, *we hypothesize that there is a continuously decaying rate of the interictal biomarker with the distance to the region where this biomarker is maximum in the case of a well sampled unifocal epilepsy* (see Fig. 1b). More formally, let a feature $f(\mathbf{X},\hat{\boldsymbol{\varphi}}_{sr})$ be a function of the distance from a spatial reference $\hat{\boldsymbol{\varphi}}_{sr}$ to all bipolar channels $\boldsymbol{\Phi} = \{\boldsymbol{\varphi}_1, \boldsymbol{\varphi}_2, \ldots, \boldsymbol{\varphi}_N\}$ at positions $\mathbf{X} \in \mathbb{R}^{N \times 3}$. We can define this decaying relationship in the form of a power law function:

$$f(\mathbf{X},\hat{\boldsymbol{\varphi}}_{sr}) = \alpha \mathrm{d}\left(\mathbf{X},\mathbf{X}_{\hat{\boldsymbol{\varphi}}_{sr}}\right)^{\kappa} \tag{1}$$

Where the distance function $\mathrm{d}(\mathbf{X},\mathbf{X}_{\hat{\boldsymbol{\varphi}}_{sr}})$ is parametrized by $\hat{\boldsymbol{\varphi}}_{sr}$. The constants $\alpha, \kappa \in \mathbb{R}$ quantify the offset of the curve from the origin and the decay of the curve. In the case of IED-$\gamma$ rates, we can observe the relationship in Fig. 1b as an example of a seizure-free case when $\hat{\boldsymbol{\varphi}}_{sr}$ is chosen to be the bipolar channel with the maximum IED-$\gamma$ rate. To test the power-law hypothesis, we apply the logarithmic operator on Eq. (1), which would result in the following change of variables:

$$\log(f(\mathbf{X},\hat{\boldsymbol{\varphi}}_{sr})) = \log(\alpha) + \kappa \log\left(\mathrm{d}\left(\mathbf{X},\mathbf{X}_{\hat{\boldsymbol{\varphi}}_{sr}}\right)\right)$$

$$\mathbf{y} = \widetilde{\alpha} + \kappa \mathbf{x}$$

Where $\mathbf{y} = \log(f(\mathbf{X},\hat{\boldsymbol{\varphi}}_{sr}))$, $\mathbf{x} = \log(\mathrm{d}(\mathbf{X},\mathbf{X}_{\hat{\boldsymbol{\varphi}}_{sr}}))$, and $\widetilde{\alpha} = \log(\alpha)$. To measure the linear relationship between $y$ and $x$, the Pearson's correlation $\rho$

will computed as follows:

$$\rho = \frac{\sum (x_i - \mu_x)(y_i - \mu_y)}{\sigma_x \sigma_y} \tag{2}$$

Where $\mu$ denotes the mean, and $\sigma$ denotes the standard deviation. Therefore, the spatial system will be characterized by computing the correlation between the feature value and its distance from the spatial reference $\hat{\boldsymbol{\varphi}}_{sr}$ in the log-log space, which is a goodness of fit of the power law function (example shown in Fig. 2).

Therefore, the spatial system in Eq. (1) will be quantified using Eq. (2). The system's response to this perturbation can therefore be quantified by the change in $\rho$, which we later define as the perturbation strength.

Since the logarithm is not well defined at zero, the channel used as the spatial reference was removed when calculating the correlation. For channels with IED-$\gamma$ rates of zero, a small value was added so that the logarithm is well defined. In this study, we used the median IED-$\gamma$ rate of all non-SOZ contacts in seizure-free patients, which we found to be 0.1 min⁻¹.

## Virtual-removal SP framework

First, we applied the perturbation by virtually removing the SOZ. We call this the virtual-removal SP framework. The first step in this framework is to record spontaneous seizures and mark the SOZ as measured by the SEEG. We then compute the IED-$\gamma$ rates for all bipolar channels. The features are then transformed into a spatial system by coupling the IED-$\gamma$ rates with their spatial distribution in relation to the SOZ. The SOZ is then virtually removed, which perturbs the coupling of the spatial system. The system's response to this perturbation is measured and used to classify the quality of the implantation of the SOZ via the surgical outcomes.

**Mathematical formulation.** In this framework, we wish to perturb the previously defined spatial system using the SOZ. We define the spatial reference $\hat{\boldsymbol{\varphi}}_{sr}$ as the channel with the maximum IED-$\gamma$ rate. To ensure that removal of the SOZ will perturb the spatial coupling, we will therefore constrain $\hat{\boldsymbol{\varphi}}_{sr}$ to be inside the SOZ, as follows:

$$\hat{\boldsymbol{\varphi}}_{sr,SOZ} = \mathrm{argmax}_{\boldsymbol{\varphi}_i \in SOZ}(f) \tag{3}$$

Hence, the spatial system is constructed using Eqs. (1) and (3) by coupling the IED-$\gamma$ rates with its Euclidean distance to the maximum IED-$\gamma$ rate channel within the SOZ (denoted as $\hat{\boldsymbol{\varphi}}_{sr,SOZ}$). More formally, the Euclidean distance between each bipolar channel $\boldsymbol{\varphi}_i$ and $\hat{\boldsymbol{\varphi}}_{sr,SOZ}$ is computed in the normalized MNI space (in mm), creating a two-dimensional space of IED-$\gamma$ rates and their distances to $\hat{\boldsymbol{\varphi}}_{sr,SOZ}$. The spatial system was characterized using Eq. (2) and is represented as $\rho_{BR}$ (i.e., before removal).

The perturbation was applied by virtually removing the SOZ, and re-computing the spatial reference:

$$\hat{\boldsymbol{\varphi}}_{sr,\overline{SOZ}} = \mathrm{argmax}_{\boldsymbol{\varphi}_i \in \overline{SOZ}}(f) \tag{4}$$

The spatial system is recomputed using Eqs. (1) and (4). This process essentially perturbs the spatial coupling, since the distances are now re-computed relative to the maximum feature channel outside the SOZ (i.e., $\hat{\boldsymbol{\varphi}}_{sr,\overline{SOZ}}$). This perturbed coupling would result in the failure of the spatial system in satisfying the power-law hypothesis, causing a reduction in the Pearson's correlation when measuring the system's response to the perturbation. The system's response is measured using Eq. (2). This is represented as $\rho_{AR}$ (after removal).

Only 10-min segments that had more than one IED-$\gamma$ per minute were considered for analysis. The correlation was set to zero for cases

when all contacts had less than one event per minute after virtually removing the SOZ (as it is noisy below that value and produces spurious correlations). The median of all valid segments was computed to remove the influence of outliers in the data.

$$\bar{\rho}_{X,i} = \mathrm{median}\left(\rho_{X,i}^{j}\right)_{j}$$

Where $X \in \{\mathrm{BR,AR,RR}\}$ (BR: Before removal; AR: After removal; RR: Random removal) for patient $i$ at segment $j$. We asked whether this decrease in $\bar{\rho}$ could be due to chance or a computational issue (since it could be inherently sample-size dependent). Therefore, non-SOZ contacts were randomly removed and $\rho_{RR}^{i}$ was re-computed for every $i^{th}$ iteration. The number of non-SOZ contacts $N_{\overline{SOZ}}$ that were randomly removed is equal to 20% the size of the SOZ ($N_{SOZ}$), which was the maximum amount for consistency across all patients, since some have $N_{\overline{SOZ}} < N_{SOZ}$. This procedure was repeated 100 times and their median is computed to produce a single $\rho_{j}$ for each segment of each patient.

**Sampling prediction.** The perturbation strength will be used to classify the patients' implantation scheme. For patient $i$, this is defined as the absolute logarithmic ratio of the correlation before removal $\rho_{BR}$ and the correlation after removal $\rho_{AR}$, as follows:

$$\hat{\rho}_{i} = \log\left(\nu + \left|\frac{\bar{\rho}_{BR,i}}{\bar{\rho}_{AR,i}}\right|\right)$$

The offset $\nu > 0$ is necessary while applying the logarithmic transformation, as the logarithm is ill-defined when $\min\left(\left|\frac{\bar{\rho}_{BR,i}}{\bar{\rho}_{AR,i}}\right|\right) = 0$. Since the logarithm is a monotonically increasing function, the choice of $\nu$ will not affect the results. The AUC was reported to demonstrate separability between the two classes (Engel IA vs. Engel IIB+). A classification threshold on the perturbation strength was determined to set an operating point on the ROC curve (see Supplementary Method S2). The optimization was performed on the MNI dataset and was tested on the CHUGA dataset.

**Interpretation.** The measured SOZ is an electrographic signature and a function of the sampling configuration. If the sampling is sparse, then the measured SOZ would either completely miss the EZ, or only incompletely overlap with it. With the proper amount of spatial sampling, the SOZ should overlap well with the EZ, therefore, suggesting that the implantation adequacy can be measured using the SOZ. Therefore, it would be important to test the sufficiency of the SOZ in perturbing the spatial system $f(\mathbf{X},\hat{\boldsymbol{\varphi}}_{sr})$. By virtually removing the measured SOZ in well sampled patients, we essentially created an SEEG implantation scheme which completely misses the 'true' SOZ, which should significantly affect the spatial coupling in Eq. (1). Virtually removing the measured SOZ in non-seizure-free patients would not necessarily result in a significant decrease in $\rho$, since the implantation may have partially or completely missed the 'true' SOZ, and therefore, the spatial coupling would not change drastically. Therefore, we hypothesized that there will be a larger decrease in $\rho$ in seizure-free patients than in non-seizure-free patients.

**Ranked SP framework**
The next step is to implement the SP framework without the need to record seizures. To do this, we constructed a spatial system similar to the one in the virtual-removal SP framework, however, without using information from the SOZ. Then, by considering each bipolar channel as a reference, we computed multiple spatial systems. This can be viewed as a series of perturbations applied to the initially constructed spatial system using each bipolar channel. This method essentially

composes all possible spatial systems, allowing us to evaluate not only the whole implantation, but all possible spatial configurations. The response to these perturbations is then measured and translated into a map by spatially ranking the strength of the system's response. Therefore, we called this the ranked SP framework. Together, this constructs the SP map, a tool that is visually interpretable without the need of recording the SOZ (Fig. 1d).

**Mathematical formulation.** The next step is to implement the SP framework without the need of recording seizures. Therefore, we defined the spatial reference $\hat{\boldsymbol{\varphi}}_{sr}$ as the channel with the maximum IED-$\gamma$ rate without any constraint to the SOZ.

$$\hat{\boldsymbol{\varphi}}_{sr} = \mathrm{argmax}_{\boldsymbol{\varphi}_{i}}(f) \tag{5}$$

The spatial system is then constructed using Eqs. (1) and (5) and is measured using Eq. (2) to obtain $\rho_{ref}$. A series of perturbations were applied to this spatial system by permuting $\hat{\varphi}_{sr}$ such that each bipolar channel $\boldsymbol{\varphi}_{i}$ is the new spatial reference:

$$\hat{\boldsymbol{\varphi}}_{sr,i} = \boldsymbol{\varphi}_{i} \tag{6}$$

The disturbed spatial system then is computed using Eqs. (1) and (6) and is measured using Eq. (2) to obtain $\rho_{i}$. This allows us to evaluate the whole implantation and all possible spatial configurations available from the implantation data. The *perturbation strength* is then computed as follows:

$$\hat{P}_{i} = 1 - (\rho_{i} - \rho_{ref}) \tag{7}$$

Therefore, each channel is associated with a perturbation strength, which quantifies the importance of a bipolar channel $\boldsymbol{\varphi}_{i}$ in constructing the spatial system in Eq. (5). Therefore, each channel is associated with a perturbation strength, which quantifies the importance of a bipolar channel $\boldsymbol{\varphi}_{i}$ in constructing the spatial system in Eq. (5). As done previously, the median of $\hat{P}_{i}$ over all segments $j \in \{1,2,\ldots\}$ is computed to remove the influence of outliers in the data:

$$\bar{P}_{i} = \mathrm{median}\left(\hat{P}_{i}^{j}\right)_{j} \tag{8}$$

**Constructing the SP map.** The next step is to visualize the perturbation indices computed in Eq. (7) for clinical interpretation of the implantation scheme. This is done by constructing a two-dimensional space which consists of each channel's perturbation strength which are ranked based on their (i) perturbation strength ($y$-axis), and (ii) Euclidean distance from each channel to the centroid of the region with significant perturbation ($x$-axis). We call this the ranked SP framework. One may interpret the process of spatially ranking the channels as a SP on a meta-system, which is the combination of all spatial systems constructed with each channel using Eq. (6). Therefore, the SP map can be considered as the response to a two-step SP.

The perturbation indices are therefore projected onto a binned two-dimensional map (i.e., the SP map). This is done by binning the distances of each bipolar channel $\boldsymbol{\varphi}_{i}$ to the centroid of the region with highest perturbation strength (denoted as $\bar{\boldsymbol{\varphi}}_{ps}$) for a total of 200 bins. The region with highest perturbation strength is defined as the set of channel coordinates with a perturbation strength greater than the 70th percentile. The centroid is computed by averaging all the coordinates in the defined region. The bipolar channels are then sorted based on their perturbation strength and their distance to $\bar{\boldsymbol{\varphi}}_{ps}$, which produces the SP map. Given that the representation will be sparse, morphological transformations were applied to the resulting image to spatially connect the perturbation indices in proximity to each other[48]. A detailed flowchart on the construction of the SP map can be found in Supplementary Fig. S11.

$\bar{\boldsymbol{\varphi}}_{ps}$ can be considered as a proxy of the SOZ, since we found that the SOZ channels have significantly larger perturbation indices than non-SOZ for both, seizure-free and non-seizure-free patients (Supplementary Fig. S9). Given that $\bar{\boldsymbol{\varphi}}_{ps}$ is a proxy of the SOZ, we indirectly used the SOZ to apply another SP. In theory, a well sampled patient should satisfy the hypothesis of a focal region of high perturbation strength and is represented by a diagonal image of decreasing intensity (as shown in Fig. 5b). A poorly sampled patient will not satisfy this hypothesis and will have an ill-defined morphology.

The SP map essentially tests all possible spatial configurations using the available implantation data, allowing one to analyze the SOZ distribution, it's perturbation profile and therefore, overall implantation quality. The overview of the ranked SP framework is shown in Fig. 5.

**Probability modeling using the SP map.** The next step is to demonstrate the clinical utility of the SP map by applying unsupervised clustering techniques to develop a probability model without using information of the patients' surgical outcomes. For interpretability, simple features were extracted from the SP map. The mean of the positive perturbation indices was computed on four quadrants (Fig. 6). The combination of features which maximizes Dunn's index[49] was selected (see Supplementary Method S3). The Dunn's index is a measure used to quantify the compactness and separability of a set of clusters (Supplementary Fig. S12). This resulted in Q1, Q2, and Q4 being chosen for the unsupervised clustering analysis. The K-means algorithm was applied to identify an optimal set of clusters without the need of ground truth labels. The elbow of the L-curve identified two main clusters independent of the information on surgical outcome. For robust centroid estimation, bootstrapping was applied by sampling the data 1000 times, and each time 75% of seizure-free and non-seizure-free patients were randomly selected and K-means was applied ($k = 2$) to obtain centroids $\mathbf{c}_1^n$ and $\mathbf{c}_2^n$ for iteration $n$. The mean of the centroids was then used in subsequent analysis, resulting in $\hat{\mathbf{c}}_1$ and $\hat{\mathbf{c}}_2$. The pipeline is shown in Supplementary Fig. S8.

A probability model was developed using features in $\hat{\mathbf{c}}_1$ and $\hat{\mathbf{c}}_2$. The Euclidean distance from $\hat{c}_1$ and $\hat{c}_2$ were computed, resulting in two vectors $\mathbf{d}_{\hat{\mathbf{c}}_1}$ and $\mathbf{d}_{\hat{\mathbf{c}}_2}$ extracted from data classified in cluster 1 and 2. The parameters of a student's t-distribution were computed to characterize the probability that a given patient in the feature space is in either cluster resulting in two distributions $T_{\hat{\mathbf{c}}_{1,2}} \sim t_\nu(\mu_{\hat{\mathbf{c}}_{1,2}}, s_{\hat{\mathbf{c}}_{1,2}}, \nu_{\hat{\mathbf{c}}_{1,2}})$.

Each patient will be characterized by their distance from both clusters (i.e., $\mathbf{d}_{\hat{\mathbf{c}}_1}$ and $\mathbf{d}_{\hat{\mathbf{c}}_2}$). The patient's $p$ values for each distribution are computed and is averaged accordingly to obtain a probability that the patient is in $\hat{\mathbf{c}}_1$, but not in $\hat{\mathbf{c}}_2$ (Supplementary Fig. S13).

**Correcting for patients with incomplete resections.** Given that a poor surgical outcome is not solely explained by poor spatial sampling, but also incomplete resections of the measured SOZ due to its proximity to the eloquent cortex as well as other anatomical constraints or palliative surgical approaches, our strategy involves applying unsupervised learning techniques to identify clusters without the need of surgical outcome information. We hypothesized that these clusters would help classify the implantation quality of the SOZ. To correct for the heterogeneities in poor outcome patients, and report 'true' classification results (i.e., good outcome = good implantation, poor outcome = poor implantation), a clinical expert (blind to the results of this study) marked whether a patient in the MNI cohort had an incomplete resection due to functional considerations. An incomplete resection was defined as a resection which did not remove the entire measured SOZ, regardless of the surgical objective, since the goal of surgery may not always be to remove the whole SOZ but also to spare the overlapping functional

cortex. Based on our definition, a patient was marked as having a complete resection if all SOZ channels were included in the resected channels (i.e., all SOZ channels were located within the resection cavity). Otherwise, if one or more channels were not included in the resected channels, the patient was marked as having an incomplete resection. Then, a data-driven approach was taken to ensure a low-likelihood of including patients with incomplete resections, for which a distribution of the resected SOZ volumes were calculated exclusively in MNI patients for training. We only considered patients above the upper quartile of this distribution (i.e., 75th percentile), and tested this threshold on the independent center dataset (i.e., CHUGA). In addition, other thresholds (70th, 80th, and 85th percentile) were employed to ensure no bias in the results.

**Surgical covariate calculations.** The SOZ, resection volumes and the resected SOZ volumes were estimated at the channel-level. Each channel of interest was inflated by a 5 mm radius sphere, and the total non-overlapping volume was numerically estimated in a 0.5 mm resolution grid. The percentage SOZ resected was estimated by first projecting each contact to a region in the MICCAI atlas[46,47]. The percentage SOZ resected is defined as follows:

$$\text{SOZ}_\% = \frac{|\text{SOZ}| \cap |\text{Resected}|}{|\text{SOZ}|}$$

Where $|\text{SOZ}|$ and $|\text{Resected}|$ denotes the cardinality of the set of regions in the SOZ and the set of regions which have been resected.

**Statistical analysis and classification measures**
The Kolmogorov-Smirnov test was used to test the normality of the data. If the data was normally distributed, we used a two-sided $t$ test and Cohen's $d$, otherwise we used a two-sided Wilcoxon's nonparametric test and Cliff's $d$ for computing effect sizes. We used paired tests wherever necessary. Correlations were tested by computing Spearman's rank correlation coefficient unless otherwise stated. All statistical tests were performed in MATLAB R2023a. The classification results were reported by first computing a confusion matrix, which is a matrix containing the number of seizure-free patients that were classified as well-sampled patients or poorly sampled patients (true positive; TP, or FN) and the number of non-seizure-free patients that are classified as poorly sampled patients or well-sampled patients (true negative; TN, or FP). The sensitivity = TP/(TP+FN) and specificity = TN/(TN+FP) were subsequently calculated to report the classification performance. The ROC curve was bootstrapped 1000 times to demonstrate statistical significance of AUC values being greater than 0.5.

**Reporting summary**
Further information on research design is available in the Nature Portfolio Reporting Summary linked to this article.

## Data availability
The clinical data that support the findings of this study were subject to ethics approval and patient consent and are available upon request if in accordance with the respective research ethics boards' policies. The raw SEEG data are protected and are not available due to data privacy laws. The processed perturbation strength data generated in this study are provided in the Source Data file. Source data are provided as a Source Data file. Source data are provided with this paper.

## Code availability
The code for detecting IED-$\gamma$ events and applying the proposed framework is available on our GitHub page[50]. Plots were produced using 'Gramm', a third-party MATLAB toolbox[51]. All data analyses were performed using MATLAB R2023a.

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

## Acknowledgements

We wish to express our gratitude to the staff and technicians at the EEG Department of the Montreal Neurological Institute and Hospital, particularly Lorraine Allard, Nicole Drouin and Chantal Lessard. This work was supported by project grants from the Canadian Institutes of Health Research (PJT-175056) and Hewitt Foundation held by B.F. K.J. and C.A. were supported by the Savoy Foundation Studentship (2021-2022). T.A. is supported by the Savoy Foundation Studentship (2021-2023). J.T. was supported by a Savoy Foundation Postdoctoral Fellowship (2021-2022). B.F. is supported by a salary award ("Chercheur-boursier clinicien Senior") from the FRQS (2021–2023).

## Author contributions

K.J., T.A., and B.F. conceptualized the project and experimental design. K.J. developed the mathematical formulation of the SP framework. K.J., T.A., A.H., J.T., C.A., D.M., P.K., S.C., J.H., L.M., C.G., J.G., and B.F. contributed to the acquisition and analysis of data. B.F., T.A., and J.G. participated in the discussion of the methodology, mathematical formulation, and results of the study. K.J., T.A., and B.F. contributed to drafting the text and preparing figures. All authors contributed to the revision of the manuscript.

## Competing interests

The authors declare no competing interests.
