## [Peer Review File · Nature Communications]

A spatial perturbation framework to validate implantation of the epileptogenic zoneREVIEWER COMMENTS

Reviewer #1 (Remarks to the Author):

In this piece of work, Jaber and colleagues propose a new framework for using interictal biomarkers in order to evaluate the success of a given sEEG electrode configuration in sampling the epileptogenic zone as part of the surgical planning in patients with drug resistant epilepsy. The study examines a problem that has high clinical impact since there is currently no iEEG-based model to evaluate whether a given implantation adequately samples the epileptogenic focus. The paper is well written and the results are interesting. However, I have some concerns in terms of methods, which should be addressed by the authors. Furthermore, referrals to previous studies in the field should be provided adequately.

Major issues:

1) It is unclear to this reviewer why a very specific biomarker of the EZ (i.e., IEDs with preceding gamma activity) was explicitly selected to test this hypothesis. Currently, as an explanation for this selection, it is provided that this biomarker has been shown to be very specific to the EZ and two references are cited, however, this is not sufficient. The researchers should examine also other seizure onset patterns (e.g., low-voltage fast activity, slow wave/DC shift followed by low-voltage fast activity, sharp theta/alpha waves, beta sharp waves, rhythmic spikes/spike-waves, and delta-brush), which occur often in the iEEG recordings [see for example work from Bartolomei's group: Lagarde et al., *Epilepsia* 2019].

The problem of insufficient sampling of iEEG has been partially tackled in previous studies [see for example work from Papadelis' group: Alhilani et al., *Clin Neurophysiol*, 2020]. These studies showed that electric source imaging performed on ictal and interictal intracranial EEG offers higher predictive value of surgical outcome compared to the conventional interpretation of iEEG data. Such kind of previous studies are very relevant and should be mentioned in the introduction section of the manuscript, while their findings should be discussed in conjunction with the presented findings here.

The IED- γ rate feature was computed using a previously described method. Although two papers are cited for the interested reader to refer to, a brief description of this method should be provided as supplementary material (text or figure). This would help understand the basics of this feature extraction that seems to be the core feature of your analysis.

There are noticeable differences in the permutation strengths between the two cohorts (see figure 4). For the MNI cohort, there is large variability for the seizure-free patients in MNI compared to the CHUGA. Please provide an explanation regarding this difference.

Figure S5(b): Based on the "knee" method, an optimal number of clusters was selected as 2 from the L-curve. Yet, in my view, the optimal number seems to be 3 or even 4 where the error becomes almost flat. Please provide a clear justification about your choice in terms of optimal number of clusters. Also, please justify why this method was used compared to other methods (e.g., elbow silhouette GAP, etc).

Figure 6 shows representative examples of one patient with good and one with poor outcome in terms of SP maps. There are striking differences in these two patterns; yet, I was wondering how consistent these patterns are across different patients. It may be worth to depict these patterns across several patients (at least 5 for each group) to highlight noticeable differences and consistency across patients.

Constructing the SP Map (Lines 636-666): This section is hard to follow as it is currently written: for instance, the description of the sorting based on two criteria (line 651: perturbation strength and distance) is unclear. This section could be enhanced through a clear flowchart diagram that describes the followed steps.

Please make sure that the referrals to your figures are correct in the text. I have noticed several occasions in the text where wrong figures were cited.

Minor issues:

There are citations in the manuscript, which refer to papers under review [line 110: Chabowsky, Klimes et al., under revision 2023]. These citations should be removed since they have not pass yet the peer-review process.

Figure 1c: Caption: c. A perturbation was applied to the spatial system by virtually removing the SOZ. -> A perturbation was applied to the spatial system by virtually removing the measured SOZ.

Line 168: in a sensitivity of 0.61 and specificity of 0.75 -> in a sensitivity of 0.61 and specificity of 0.75, respectively.

In the caption of Figure 4, it is stated that "a. MNI cohort has an AUC of 0.75, and CHUGA AUC of 0.8". Yet, these values are displayed in panels b and c, not a.

Line 215,217: Referral to Figure 7 is inappropriate since the figure does not display the sensitivity and specificity of surgical outcome classification.

Line 491: It is stated that new bipolar channel coordinates were computed as midpoints. Does the y-axis on Figure 6 represents the bipolar channels or the original ones?

Line 504: The terms "perturbation", "spatial system", and "response" are used interchangeably in the text. A clear explanation/statement about what these terms represent it would be helpful.

Supplementary Figure S1: It is stated that out of 75 patients, 50 were eventually selected (40+10 in the flowchart), while 22 were excluded. Please clarify. Also, in the flowchart CHUGA has 42 patients while in text (line 441) there are 40 which are indicated for the CHUGA. Maybe add a subtraction symbol in the flowchart to make it meaningful. I am not sure whether I am missing something here.

Reviewer #2 (Remarks to the Author):

This manuscript ventures into a refreshing new direction in the epilepsy literature, by proposing an interesting and well-developed model using electrophysiological features alone, whereby one may rate or 'score' the adequacy (or quality) of an SEEG implantation procedure. The model combines SEEG channel-specific interictal biomarkers (interictal epileptiform discharges with preceding gamma activity, IED- γ) and their exponential decay rates to construct a spatial manifold capturing the dynamics of the epileptic system. Subsequently, the authors assess the impact on network coupling by delivering a virtual perturbation [simulated via removal of the seizure-onset zone (SOZ) contacts], and use the strength of the perturbation effect to classify the quality of the implantation scheme for its sampling of the SOZ. The result is a spatial perturbation map that serves as a useful tool for clinicians and scientists to grade the quality of an SEEG study, and the model is validated through several permutations of data analysis linked to clinical outcomes, with nice descriptions of case examples.

The model uses the appropriate level and scale of patient data, collected across two well-established and renowned epilepsy centers, using stringent inclusion criteria and suitable processing methods. This Reviewer appreciated the decision to exclude a priori, the more ambiguous Engel IB-IIA group of patients. The demographics are comparable between the MNI and the CHUGA cohorts, suggesting homogeneity between the two patient populations. The SEEG-derived features are firmly established in the literature, and the manuscript presents a sound and rigorous mathematical/computational approach, bolstered by validation of the data using bootstrapping methods and supplementary information. The latter includes excellent and representative case examples, highlighting the clinical utility and impact of the study. Assumptions or data thresholds are appropriately validated, either quantitatively or through references made in the literature. Clinical experts were used at appropriate time points, such as using a blind reviewer to assess whether a resection was incomplete due to functional considerations. The authors made

appropriate corrections to their data processing pipeline, including the correction for patients with insufficient resection, once again applying a more stringent limitation to their data. The integration of probability modelling and unsupervised clustering (i.e. machine learning) techniques adds further to the novelty of the project, and to the best of this Reviewer's knowledge, these are utilized in straight-forward fashion staying consistent with good practices in the literature. The figures are clear, legends are concise, and readability and overall presentation are excellent.

The signal processing and analytical methods used in the manuscript are comprehensive and appropriately employed, applying well-established channel-specific signal properties to a unique inception of a spatial manifold system, whose perturbation is useful in providing qualitative interpretation and classification of the overall SEEG implantation. This data-driven approach is equation-free and independent of other potentially confounding factors, built up from the electrophysiological properties inherent to the channel recordings themselves. The analytical approach is nicely described, and easy to follow, in a step-wise fashion. The statistical analyses appear sound and are well-described. The data interpretation and conclusions drawn are validated by the results presented, in addition to the wealth of supplementary information provided.

The manuscript is clear, easily understood and nicely organized. In fact, the step-wise progression of analytics is relatively easy to follow. Terms are nicely explained throughout the manuscript. The context is well-developed with useful and appropriate references, which serve to validate the methodological decision-points in the analytical workflow of this project. The manuscript cements its inception and utility in the broader context of the world of SEEG and epileptogenic brain networks, and has real-world applications to the field of epilepsy and epilepsy surgery. I commend the authors on an excellent and well-composed manuscript here.

Suggested Improvements:

This Reviewer is appreciative for the chance to provide feedback, in the hopes of further improving the manuscript. Some of these suggestions are conceptual, and some are stylistic, as follows:

1) Page 3, Lines 45-46 – There is some discussion in the epilepsy community over the definition of the Epileptogenic Zone (EZ). Here, the authors employed one common definition, i.e. 'the minimum amount of cortex needed to be removed to achieve seizure freedom.' However, I would strongly recommend including the alternate, network-based definition from Talairach and Bancaud (who invented the SEEG procedure), whereby the EZ is defined as the network or region of earliest seizure organization and propagation. This would perhaps better underscore emphasis on 'networks' rather than a singular 'focus,' and is a better, more accurate historical definition to use for the analytical methods described in this manuscript, which seek to evaluate the quality of SEEG implantations. Perhaps a reference to the original work by Talairach and Bancaud might be fitting here, also.

2) Page 3, Line 49 – "... it is limited by the fact that only a restricted brain region can be explored..."
SEEG is an invasive methodology that allows for exploration of brain 'networks,' and is not confined to a restricted brain region in its sampling of those networks. Rather, to clarify this sentence, I would amend to something like '... it is limited by the constraint of implanting a finite number of electrodes ...'

3) Page 3, Line 56 – The spelling of the word 'miss-identification' should be corrected to 'misidentification.'

4) Page 4, Lines 79-80 – The reference formatting for 'interictal biomarkers' needs correction here, I believe. Are the authors referring to refs 1 and 11? It is written as 111.

5) Page 4, Line 88 – The spelling of 'persued' should be corrected, to 'pursued.'

6) Page 8, Lines 168-169 – It is interesting that no effect or difference was observed between MRI-positive and MRI-negative patients in the two cohorts. It is counterintuitive, in that one would expect a better SEEG implantation and subsequent increased perturbation effect following virtual

removal of the SOZ, in patients with MRI-positive findings. Perhaps the authors could include a short statement to explain their observation and possible justification, later in the discussion. Also, did the study perhaps include patients who underwent morphometric analytical post-processing (MAP)? Just curious, and if so, it would be interesting to clarify this point further.

7) Page 11, Lines 228-229 – It is curious that a high fraction of patients (nearly 62%) were deemed as 'incomplete resections.' This seems high, and it would be useful for the authors to clarify how this was determined. Was the surgeon considering the resection as 'incomplete' immediately post-operatively (i.e. surgical goal not achieved); was the resection intentionally limited owing to overlapping functional networks (in which case, the surgical goal may still have been achieved, and considered 'complete'); or is it based on recurrence of seizures, leading to repeat imaging, in which case the surgical target was not resected (again, 'surgical goal not achieved')? Some further clarification will be essential here.

8) Page 21, Line 465 – The reference (300) at the end of the sentence, should be corrected (30?).

9) Page 39, Fig. 1a (right figure, brain illustration with electrode localization) – The schematic illustrates electrodes penetrating the brain, and clearly shows the concept of contact distances used in the analyses. However, these electrodes appear to be unconventional and may be exaggerated in placement, for the effect of explaining the analysis. Do the surgeons at these centers implant orthogonally? Obliquely? The figure shows a rather haphazard electrode implantation, and so perhaps some clarification in the body of the text will be helpful. And importantly, if this is a true implantation schematic, the authors may disregard this comment, as surgical practices do vary for this technique.

Respectfully submitted,
Demitre Serletis, MD PhD FRCSC FAANS FAES
Epilepsy Center, Neurological Institute, Cleveland Clinic, USA

Reviewer #3 (Remarks to the Author):

Code Revision Notes:

The code files were very well organized with clear instructions on how to install, run, and replicate results of the manuscript titled "A Spatial Perturbation Framework to Validate Implantation of the Epileptogenic Zone". The code also contained description of the script, inputs, and outputs as well as commented lines to help the users modify and implement their own tests. A readme file contained instructions to install and use the code which made it easy to install, run, and visualize results. A license agreement and pseudocode were also provided with the code. Two sample EEG data files were also included to test the code (one for well sampled case and one for poorly sampled case). A breakdown of the revision and some recommendations are listed below:

- The code is well written, well described and commented whenever necessary except in the four scripts (boxplot_gramm.m, plotFeatureSpace.m, Scatter_gramm.m, computeSpikeGamma.m) which lacked any description or comments. For completion and interpretability, these scripts need to be described and commented like the others.
- The code was revised and tested (all functions and scripts were tested one by one). All codes ran smoothly without errors and generated the expected results of the manuscript and as described in the readme file.
- The methodology described in the manuscript was perfectly replicated by the code and the pseudocode describes the methodology well.
- The manuscript had some ambiguities in terms of steps (specifically line 636 - 666 "Constructing the SP Map" section on page 29 which was hard to understand or follow since the steps were not

clearly described mathematically), but the code and the pseudocode helped clarify them all. The authors should consider enhancing that paragraph in the manuscript.

- Some typos in the comments within the code scripts were found, the authors may consider correcting them (such as line 127 in computeSPMap.m ["feautres" features] among others).
- The methodology seems to work with any feature extracted from iEEG. The authors may consider mentioning this fact in the discussion highlighting its use and limitations when used with other features if any.
- Overall, this work is a very well written, well coded, and very useful as a methodology to discriminate between well-sampled and poorly sampled iEEG implantations.

Response to Reviews

Details of Submission:

- Manuscript Number: NCOMMS-23-55758A
 - Title: A Spatial Perturbation Framework to Validate Implantation of the Epileptogenic Zone
-

Answers to Reviewer 1's comments

In this piece of work, Jaber and colleagues propose a new framework for using interictal biomarkers in order to evaluate the success of a given sEEG electrode configuration in sampling the epileptogenic zone as part of the surgical planning in patients with drug resistant epilepsy. The study examines a problem that has high clinical impact since there is currently no iEEG-based model to evaluate whether a given implantation adequately samples the epileptogenic focus. The paper is well written and the results are interesting. However, I have some concerns in terms of methods, which should be addressed by the authors. Furthermore, referrals to previous studies in the field should be provided adequately.

Major issues:

1) It is unclear to this reviewer why a very specific biomarker of the EZ (i.e., IEDs with preceding gamma activity) was explicitly selected to test this hypothesis. Currently, as an explanation for this selection, it is provided that this biomarker has been shown to be very specific to the EZ and two references are cited, however, this is not sufficient. The researchers should examine also other seizure onset patterns (e.g., low-voltage fast activity, slow wave/DC shift followed by low-voltage fast activity, sharp theta/alpha waves, beta sharp waves, rhythmic spikes/spike-waves, and delta-brush), which occur often in the iEEG recordings [see for example work from Bartolomei's group: Lagarde et al., *Epilepsia* 2019].

Response:

*We would like to thank the Reviewer for their comment. There may be some misunderstanding regarding the feature we used. We truly appreciate you giving us a chance to clarify this point, as it will help improve the foundation of our manuscript. The biomarker we used for this work is based on the interictal and not the ictal EEG. Indeed, the ultimate goal of this study was to develop a model that uses only interictal information to infer implantation adequacy of the seizure-onset zone (SOZ) that can be deployed directly after electrode implantation. Therefore, we opted to investigate an interictal biomarker that has been shown independently by two groups to localize the EZ with high specificity, as we wish to quantify the spatial distribution of the 'true' SOZ, and not the measured SOZ^{1,2}. IED- γ as used in this study is defined as an IED with significant gamma activity preceding its onset in the **interictal** EEG. This is different from the pre-/peri-ictal spiking that may occur prior to the onset of seizures as one of several seizure onset patterns as rightfully pointed out by the Reviewer. The following modification has been done to address this potential confusion.*

On revised Manuscript Page 5-6, Paragraph 2, Lines 94-102, under “Introduction”:

“...to measure the success of a given SEEG electrode configuration in sampling the EZ as part of pre-surgical planning. ~~Here, we opted to leverage on interictal epileptiform discharges (IEDs) with preceding gamma activity (30-100 Hz; IED- γ), as this marker has been shown to be highly specific to the EZ^{12,24}.~~ The use of interictal biomarkers for the development of the framework can enable one to evaluate the SEEG configuration without the need to record seizures. Therefore, we opted to only consider interictal biomarkers in this study since our ultimate goal was to perform a seizure-independent evaluation of the SEEG configuration. More specifically, we considered IEDs with preceding gamma activity (30-100 Hz; IED- γ), as this interictal marker has been shown to be highly specific to the EZ^{12,21}.”

However, the Reviewer’s comment raises a very important question regarding the use of IED- γ activity for the development of our framework. Therefore, to further strengthen our choice of using IED- γ rates, we have included additional analyses using more traditional interictal markers such as IEDs and ripple (80-250 Hz) rates that are found in the Supplementary Material. We did not use fast ripples (250-500 Hz), as the sampling rate of the test set was limited to only 512 Hz. Indeed, we were able to show that the IED- γ rate performs best in classifying the implantation scheme. The comparative results are provided in the Supplementary Material and are referred to in the Discussion section of the manuscript.

On revised Manuscript highlight Page 19-20, Paragraph 3, Lines 419-428, under “Discussion”:

“To strengthen the foundation of our framework on the use of IED- γ activity, we investigated the virtual-removal SP framework using more traditional interictal biomarkers such as IEDs and ripples (80-250Hz) (see **Supplementary Fig. S10**). In theory, the framework can be applied to any interictal marker with the requirement that it is specific to the EZ. We found that IEDs constructed a sub-optimal spatial system given that they are rather unspecific to the EZ¹⁶. Virtually removing the SOZ perturbs the IED spatial system less than in the IED- γ spatial system in both MNI and CHUGA patient cohorts. We also found that ripples perform extremely poorly in the virtual-removal SP framework. The reason for the poor perturbation profile of the ripple-rate spatial system could be due to the confound of physiological ripples, given large spatial variability across different brain regions^{26,27}.”

The Figure from the Supplementary is inserted below for reference.

Perturbation strengths

Supplementary Figure S10: Perturbation strength of IED- γ predicts surgical outcome. The perturbation strength is computed for each patient as well as different interictal EZ biomarkers (described in **Methods**). **(a)** IED rates achieve near significance ($\hat{\rho}_{SF}, \hat{\rho}_{non-SF} = 1.30 (0.37), 1.15 (0.29)$; $p = 0.054$; $d = 0.34$; $AUC = 0.67$ (95% CI: 0.51, 0.87)), IED- γ rates achieve significance with a moderate effect ($\hat{\rho}_{SF}, \hat{\rho}_{non-SF} = 1.40 (3.95), 1.14 (0.36)$; $p < 0.01$; $d = 0.50$; $AUC = 0.75$ (95% CI: 0.62, 0.94)), however, ripple rates do not produce an SOZ network that evaluates the measured SOZ ($\hat{\rho}_{SF}, \hat{\rho}_{non-SF} = 1.15 (0.35), 1.06 (0.61)$; $p = 0.98$; $d = 0.005$; $AUC = 0.50$ (95% CI: 0.32, 0.71)). **(b)** The ROC curves of perturbation strengths using IEDs, IED- γ , and ripples. The threshold used to classify surgical outcome is obtained from the ROC curve of the IED- γ spatial system. **(c)** We found the same trends in the CHUGA patient cohort, with IED rates now achieving significance; ripples remained

non-significant (**IED**: $\hat{\rho}_{SF}, \hat{\rho}_{non-SF} = 1.20 (0.35), 0.97 (0.29)$; $p < 0.05$; $d = 0.53$; $AUC = 0.76$ (95% $CI: 0.56, 1.00$), **IED- γ** : $\hat{\rho}_{SF}, \hat{\rho}_{non-SF} = 1.28 (0.56), 0.99 (0.31)$; $p < 0.05$; $d = 0.60$; $AUC = 0.80$ (95% $CI: 0.62, 1.00$), **Ripples**: $\hat{\rho}_{SF}, \hat{\rho}_{non-SF} = 1.19 (0.43), 1.03 (0.98)$; $p = 0.80$; $AUC = 0.47$ (95% $CI: 0.11, 0.85$)). In both figures, non-SF and SF patients are shown as red and green dots, respectively. MRI-negative patients are represented as a circle, and MRI-positive are presented as a square. Summary statistics are represented as median (IQR). \tilde{p}^* is tested on the CHUGA cohort, obtaining a sensitivity of 0.61 and specificity of 0.75. **(d)** The ROC curves of the IED, IED- γ and ripple spatial systems in the CHUGA cohort. The threshold is visualized as a horizontal dotted line in **(a)** and **(c)**. Abbreviations: IED=interictal epileptiform discharge, MNI=Montreal Neurological Institute, CHUGA=Grenoble Alpes University Hospital Center, IED=interictal epileptiform discharge, IQR=interquartile range, ROC=receiver operating characteristics. Statistical significance shown in asterisks: * $p < 0.05$, ** $p < 0.01$

We also thank the Reviewer for bringing to our attention the important paper from the Marseille group regarding the various seizure-onset patterns. We included it regarding the definition of the SOZ.

On revised Manuscript highlight Page 24, Paragraph 3, Line 539-540, under “Methods”:

“...in the EEG at seizure onset independent of the fast activity content and seizure-onset pattern^{28,38}.”

The problem of insufficient sampling of iEEG has been partially tackled in previous studies [see for example work from Papadelis’ group: Alhilani et al., Clin Neurophysiol, 2020]. These studies showed that electric source imaging performed on ictal and interictal intracranial EEG offers higher predictive value of surgical outcome compared to the conventional interpretation of iEEG data. Such kind of previous studies are very relevant and should be mentioned in the introduction section of the manuscript, while their findings should be discussed in conjunction with the presented findings here.

Response:

We thank the Reviewer for their comment and acknowledge that the work from Papadelis’ group indeed partially tackles the problem of spatial sampling from a different angle. We did not mention their study in the manuscript since they did not directly evaluate the SEEG implantation of the EZ. We included literature where the study directly aimed to solve the problem of the sub-optimal spatial sampling of the EZ. However, the Reviewer raised a very important piece of literature that needs mentioning, as the methodology could be used to directly evaluate the implantation accuracy. Indeed, we had no intention of ignoring any literature surrounding the topic of spatial sampling, and have now included their methodology and results in the introduction and discussion sections of the paper.

On revised Manuscript highlight Page 5, Paragraph 1, Line 83-90, under “Introduction”:

“It showed a predominant outward information flow from the spike onset in seizure-free patients. This might suggest that good EZ coverage is required to observe this flow, but this was not assessed in this work. Other studies have applied electrical source imaging methods on SEEG data to improve the localization of the EZ¹⁸, or predict surgical outcomes^{19,20}. They have shown that localizing interictal epileptiform discharges (IEDs) or seizures recorded by the SEEG can localize far-field activity not sampled by the SEEG, and using this information, predict surgical outcomes. However, they did not develop a model to score the adequate sampling of the EZ by a given electrode configuration. Therefore, ~~to the best of our knowledge, there is no research~~ there is a need to develop which proposes a simple model based on SEEG alone to evaluate whether a given implantation scheme has adequately sampled the epileptic focus.”

On revised Manuscript highlight Page 21, Paragraph 2, Line 456-462, under “Discussion”:

“Previous studies have only attempted to refine the localization of the epileptic focus^{14,29}, or aimed to find spiking regions or SOZ missed by the SEEG^{15,18-20}. Although finding novel spiking regions is a valuable tool to study the interictal network of spikes, they are also known to delineate the irritative zone, which encompasses a larger region than the EZ¹⁶. In addition, while delineating seizure-onset regions missed by the SEEG can play an important role in determining whether the electrode configuration has sampled the ‘true’ SOZ, this alone does not provide prospective value of whether the implantation was sufficient for delineating and surgically removing the measured SOZ to achieve seizure-freedom.”

The IED- γ rate feature was computed using a previously described method. Although two papers are cited for the interested reader to refer to, a brief description of this method should be provided as supplementary material (text or figure). This would help understand the basics of this feature extraction that seems to be the core feature of your analysis.

Response:

Thank you for providing us with the opportunity to clarify this method further. A brief explanation of the feature extraction process is now added in Supplementary Method S1:

On revised Supplementary highlight Page 2, Paragraph 1-2, Line 22-45, under “Supplementary Method S1”:

IED and ripple rates were previously shown to localize the EZ and were used to validate the use of the IED- γ rate¹⁻³. IEDs and ripples were automatically detected using previously validated detectors^{4,5}. The IED rates, IED- γ , and ripple rates were then computed and reported as rates per minute. Fast ripples (250-500Hz) were not

considered, as the validation cohort mainly contained SEEG data with sampling rate of 512 Hz.

The IED- γ detection was employed in accordance with Thomas et al.'s publication³ using the code available at the GitHub repository: <https://github.com/Lab-Frauscher/Spike-Gamma>. The following are the key steps:

1. Detect the IEDs using the Janca detector⁴: Firstly, the signal was downsampled to 200Hz, and the envelope of the bandpassed signal [10,60Hz] is computed. An IED is detected within a 5s moving window (80% overlap) if the upper percentile (i.e., $3.65 \times (\text{mode} + \text{median})$) of the distribution of the signal envelope computed from the 5s moving window, intersects with the signal envelope. The IED detections using the Janca detector were validated in previous studies^{3,6,7}.
2. Postprocessing: All detected IEDs that occurred within 300 ms in the same channel were ignored. IEDs were also ignored if the detections were made simultaneously in at least 50% of channels.
3. IED- γ detections: For each detected event, the IED onset and offsets were delineated after the signal was band-passed [0.3, 500Hz] for MNI data, and [0.3, 250Hz] for CHUGA (sampling rates of 2000 and 512 Hz, respectively) using a fourth order Butterworth filter. After the onset and offset was determined, the γ -activity [30-100Hz] is compared before the IED onset and after the IED offset using a 500ms window. An IED is classified as an IED- γ if the 500ms window preceding the onset of the IED contains significant γ -activity greater than two standard deviations with at least 3 cycles.

There are noticeable differences in the permutation strengths between the two cohorts (see figure 4). For the MNI cohort, there is large variability for the seizure-free patients in MNI compared to the CHUGA. Please provide an explanation regarding this difference.

Response:

We thank the Reviewer for their careful reading of the manuscript and our Figure. Indeed, this Figure demonstrates a larger variability in the MNI cohort than the CHUGA cohort. This was mainly driven by five patients who skewed the box plot's upper quartile. In fact, one outlier also exists in the CHUGA dataset, but it is less apparent as one outlier is not sufficient to widen the interquartile ranges in the boxplot. Nevertheless, this is very important, and we thank the Reviewer for pointing it out.

*We conducted a careful examination of these individual patients in relation to the rest of the cohort to determine which factor might have caused this change. We found that the reason for all these patients being outliers in both the MNI (n=5) and the CHUGA datasets (n=1) is driven by a steep reduction in the IED- γ rates after virtually removing the SOZ compared to before its removal, as seen with very focal and well implanted SOZs. Indeed, these patients experienced a median reduction of 95% [86-98] in the MNI cohort, and the patient in the CHUGA cohort exhibited a reduction of 85%. This was enough to drive the upper quartile more upwards in the MNI cohort boxplot shown in **Figure 4**. After careful analysis, we believe that these outliers in perturbation strength are the results of very*

successful implantations which sharply reduced IED- γ rates after virtually removing the measured SOZ, and are not an artifactual result of the SP framework.

Figure S5(b): Based on the “knee” method, an optimal number of clusters was selected as 2 from the L-curve. Yet, in my view, the optimal number seems to be 3 or even 4 where the error becomes almost flat. Please provide a clear justification about your choice in terms of optimal number of clusters. Also, please justify why this method was used compared to other methods (e.g., elbow silhouette GAP, etc).

Response:

We thank the Reviewer for pointing out this issue. We agree that 3 or 4 flattens the error, however, to prevent any subjectivity in deciding the knee-point, we used a third-party MATLAB function called `knee_pt` as mentioned in the Supplementary Material. Please see the following citation

Dmitry Kaplan (2024). Knee Point (<https://www.mathworks.com/matlabcentral/fileexchange/35094-knee-point>), MATLAB Central File Exchange. Retrieved February 26, 2024.

We used a very simple error function for calculating the L-curve, defined as the sum of distances of each point in cluster i to centroid i . Unfortunately, the silhouette error function is not defined for $k = 1$, so it is not possible to determine whether the elbow is at $k = 2$. Given that the existence of two clusters was our a-priori hypothesis, since we expect to classify good and poor sampling (i.e., two clusters), we opted to look into a more objective way of proving that the clusters indeed support this hypothesis. Using the MATLAB function `evalclusters`, we find that the optimal number of clusters is 2. The code snippet can be found below:

```
evaluation = evalclusters(X_mni,"kmeans","silhouette","KList",1:10);
```

```
evaluation =
```

```
SilhouetteEvaluation with properties:
```

```
NumObservations: 50
```

```
InspectedK: [1 2 3 4 5 6 7 8 9 10]
```

```
CriterionValues: [NaN 0.7117 0.6575 0.5943 0.5536 0.4304 0.4481 0.4702 0.4553 ...
```

```
]
```

```
Optimalk: 2
```

As suggested by the Reviewer, we also analyzed the GAP statistic, and found that the optimal number of clusters using the MATLAB function `evalclusters` is also two. The code snippet can be found below:

```
evaluation =  
evalclusters(X_mni, "kmeans", "GAP", "KList", 1:10, 'SearchMethod', 'firstMaxSE');
```

evaluation =

GapEvaluation with properties:

```
NumObservations: 50  
  InspectedK: [1 2 3 4 5 6 7 8 9 10]  
 CriterionValues: [0.1348 0.2779 0.3583 0.4079 0.4060 0.3602 0.3998 0.3985 ... ]  
  OptimalkK: 2
```

We have added the following in the revised Supplementary methods highlight Page 4, Paragraph 4, Lines 93-94

“...as the optimal number of clusters (see **Supplementary Fig. S5b S8b**). We also determined the optimal number of clusters using the GAP statistic¹⁰, and found that $k_{opt} = 2$.”

Figure 6 shows representative examples of one patient with good and one with poor outcome in terms of SP maps. There are striking differences in these two patterns; yet, I was wondering how consistent these patterns are across different patients. It may be worth to depict these patterns across several patients (at least 5 for each group) to highlight noticeable differences and consistency across patients.

Response:

*We agree with the Reviewer’s comment that more examples would benefit the reader. Therefore, as per the Reviewer’s suggestion, we have now provided examples of five seizure-free and five non-seizure-free patients from the MNI and CHUGA patient cohorts in the Supplementary Materials on page 8-11 (**Supplementary Fig. S4-7**) to demonstrate the consistency across patients.*

We have also referenced the Figure in the revised Manuscript as follows:

On revised Manuscript highlight Page 12, Paragraph 1, Line 240-241 under “Results”

*“...that the SOZ may have been missed. Indeed, these patterns are consistent across patients from MNI and CHUGA as shown in **Supplementary Fig. S4-7.**”*

The figures are shown below:

MNI Seizure-free

Supplementary Figure S4: Examples of seizure-free SP maps from the MNI patient cohort. Five examples of the SP maps computed from seizure-free patients in the MNI cohort are shown to demonstrate the consistencies in the patterns described in **Figure 6** of the main manuscript.

Abbreviations: MNI=Montreal Neurological Institute, Q=Quadrant, SP=Spatial perturbation.

MNI
Non-seizure-free

Supplementary Figure S5: Examples of non-seizure-free SP maps from the MNI patient cohort. Five examples of the SP maps computed from non-seizure-free patients in the MNI cohort are shown to demonstrate the consistencies in the patterns described in **Figure 6** of the main manuscript. Abbreviations: MNI=Montreal Neurological Institute, Q=Quadrant, SP=Spatial perturbation.

CHUGA

Seizure-free

Supplementary Figure S6: Examples of seizure-free SP maps from the CHUGA patient cohort. Five examples of the SP maps computed from seizure-free patients in the CHUGA cohort are shown to demonstrate the consistencies in the patterns described in **Figure 6** of the main manuscript, when applying the method on a different patient cohort extracted from an independent center. Abbreviations: CHUGA= Grenoble Alpes University Hospital Center, Q=Quadrant, SP=Spatial perturbation.

CHUGA

Non-seizure-free

Supplementary Figure S7: Examples of non-seizure-free SP maps from the CHUGA patient cohort. Five examples of the SP maps computed from non-seizure-free patients in the CHUGA cohort are shown to demonstrate the consistencies in the patterns described in **Figure 6** of the main manuscript, when applying the method on a different patient cohort extracted from an independent center.

Abbreviations: CHUGA= Grenoble Alpes University Hospital Center, Q=Quadrant, SP=Spatial perturbation.

Constructing the SP Map (Lines 636-666): This section is hard to follow as it is currently written: for instance, the description of the sorting based on two criteria (line 651: perturbation strength and distance) is unclear. This section could be enhanced through a clear flowchart diagram that describes the followed steps.

Response:

We thank the Reviewer for their careful attention to the details of the methodology proposed in the manuscript, and for providing us with the opportunity to further make clarifications which can improve the clarity and quality of our manuscript. We agree that

the paragraph lacks precision in language and have prepared the suggested flowchart which we hope will now add more clarity regarding how the ranking is performed to construct the SP map. We have inserted the following figure in the revised Supplementary Materials, and have referred to this Figure in the Manuscript accordingly.

Supplementary Figure S11: Detailed flowchart of constructing the SP map. Firstly, a spatial system is constructed using the maximum IED- γ channel as the spatial reference (right column). This spatial system is perturbed by permuting the spatial reference with each channel (left column). The perturbation strength is computed as the change in the spatial system after applying a permutation with channel i . This is done for all channels available in the SEEG implantation scheme, which produces a distribution of perturbation

strengths, where each perturbation strength is associated with a channel. A second-step perturbation is applied by redefining the spatial reference as the region of high perturbation strength. The spatial reference is computed by averaging the channel coordinates which have perturbation strengths larger than the 70th percentile of the distribution. This is used as the *x*-axis of the SP map. The channels are sorted in ascending order and plotted in the *y*-axis. A uniform bin size of 200 is used in the *x*-axis for uniformity across patients.

The figure is also referenced in the following paragraph:

On revised Manuscript highlight Page 33, Paragraph 2, Line 734-735 under “Methods”

“...spatially connect the perturbation indices in proximity to each other⁴⁶. A detailed flowchart on the construction of the SP map can be found in Supplementary Fig. S11.”

Please make sure that the referrals to your figures are correct in the text. I have noticed several occasions in the text where wrong figures were cited.

Response:

Thank you. We apologize for this oversight. We fixed the following referrals:

- On revised Manuscript highlight Page 11, Paragraph 1, Lines 208-210 under “Results”

“...as shown in **Supplementary Fig. S5a S3a**. The perturbation strength **is was** also not significantly correlated to the number of days between the two nights ($p = 0.38$; **Supplementary Fig. S5b S3b**).”

- On revised Manuscript highlight Figure 8, we swapped Figure 8a and 8b to fix the order of referrals after correcting for the wrong citations. We also corrected the Figure legend accordingly:

“Figure 8: Poor implantations record larger SOZ: The SOZ volumes were estimated as described in **Methods**. ~~(a) The SOZ volumes are the same for seizure-free and non-seizure-free patients classified as well sampled.~~ **(b) (a)** The SOZ volume tends to be larger when comparing the patients in cluster **12** (~~well-sampled~~ **poorly-sampled**; median volume = **6.3 (6.9) cm³**) **and** with the patients in cluster **21** (~~poorly-sampled~~ **well-sampled**; median volume = **4.2 (2.2) cm³**), although not statistically significant. **(b)** The SOZ volumes are not significantly different for seizure-free (median volume = **4.1 (2.8) cm³**) and non-seizure-free patients (median volume = **4.2 (2.1) cm³**) classified as well sampled. After correcting for insufficient resections (i.e., removing patients with resected SOZ volume below **3.3 cm³**), patients classified as poorly sampled have a significantly

larger SOZ volume (median volume= 8.8 (6.1) cm³) than well-sampled patients (median volume= 5.1 (2.2) cm³) in the (c) MNI cohort, with the same trend in the (d) CHUGA cohort ($\hat{\epsilon}_2$: 9.7 (7.2) cm³; $\hat{\epsilon}_1$: 5.8 (2.4) cm³). However, this is (not significant due to low statistical power). The center line of the boxplot represents the median, and the box limits represent the 25th and 75th percentile. The whiskers represent the complete range of values that are within 1.5 times the interquartile range. The notch around the median center line of the boxpot represents the 95% confidence interval of the median. The summary statistics are reported as median (IQR). Abbreviations: IQR=interquartile range, TP = true positive (well-sampled seizure-free patients), FP = false positive (well sampled non-seizure-free patients), TN = true negative (poorly sampled non-seizure-free patients), SOZ = seizure onset zone. Statistical significance is shown in asterisks: * $p < 0.05$.”

- On revised Manuscript highlight Page 16, Paragraph 1, Line 325 under “Results”

The model classified two sub-groups within the group of seizure-free patients: seizure-free patients classified as well-sampled, and seizure-free patients classified as poorly sampled (Figure 9a, b).
- On revised Manuscript Page 33, Paragraph 3, Line 738 under “Methods”

“...seizure-free and non-seizure-free patients (Supplementary Fig. S7 S9).”
- On revised Manuscript highlight Page 34, Paragraph 2, Line 747 under “Methods”

The overview of the ranked SP framework is shown in Figure 4 Figure 5.
- On revised Manuscript highlight Page 34, Paragraph 3, Line 753 under “Methods”

“The combination of features which maximizes Dunn’s index⁴⁷ was selected (see Supplementary Methods S2 S3).”
- On revised Manuscript highlight Page 25, Paragraph 2, Line 562 under “Methods”

“An example output of the algorithm is shown in Supplementary Fig. S2ba.”
- On revised Manuscript highlight Page 27, Paragraph 1, Line 596 under “Methods”

“...we hypothesize that there is a continuously decaying rate of the interictal biomarker with the distance to the region where this biomarker is

maximum in the case of a well sampled unifocal epilepsy (see **Figure 1b**).”

Minor issues:

There are citations in the manuscript, which refer to papers under review [line 110: Chabowsky, Klimes et al., under revision 2023]. These citations should be removed since they have not pass yet the peer-review process.

Response:

We thank the Reviewer for their comment. The paper is now published in *Clinical Neurophysiology*. Citation is updated on revised Manuscript highlight Page 7 and 24, Paragraph 1, Line 130 and 523-524, under “Introduction” and “Methods”:

“...as commonly observed after seizures when lowering of antiseizure medication²², are more accurate in predicting the EZ (~~Chybowski, Klimes et al., in revision 2023~~)²³”

“...usually IEDs are higher when seizures have been recorded (~~Chybowski, Klimes et al., in revision 2023~~)²³.”

Bartłomiej Chybowski, Petr Klimes, Jan Cimbalnik, Vojtech Travnicek, Petr Nejedly, Martin Pail, Laure Peter-Derex, Jeff Hall, François Dubeau, Pavel Jurak, Milan Brazdil, Birgit Frauscher. Timing matters for accurate identification of the epileptogenic zone, *Clinical Neurophysiology*, Volume 161, 2024, Pages 1-9, ISSN 1388-2457.

Figure 1c: Caption: c. A perturbation was applied to the spatial system by virtually removing the SOZ. -> A perturbation was applied to the spatial system by virtually removing the measured SOZ.

Response:

Thank you. We have now made the correction on the revised Manuscript highlight, Figure 1c caption

“**c** A perturbation was applied to the spatial system by virtually removing the **measured** SOZ.”

Line 168: in a sensitivity of 0.61 and specificity of 0.75 -> in a sensitivity of 0.61 and specificity of 0.75, respectively.

Response:

Thank you. This is now corrected. See revised Manuscript highlight, Page 10, Paragraph 2, Line 199

“...then tested on the CHUGA cohort for IED- γ prediction scores, resulting in a sensitivity of 0.61 and specificity of 0.75, respectively.”

In the caption of Figure 4, it is stated that “a. MNI cohort has an AUC of 0.75, and CHUGA AUC of 0.8”. Yet, these values are displayed in panels b and c, not a.

Response:

We thank the Reviewer for their careful attention. We have changed the figure legend accordingly (changes shown in highlight). We have also made additional changes to include more statistics as per Nature’s checklist on statistical reporting.

“Figure 4: Perturbation strength predicts surgical outcome. The perturbation strength is computed for each patient (described in **Methods**). **a** ~~The MNI cohort has an AUC of 0.75, and the CHUGA cohort has an AUC of 0.80.~~ In both centers, we see that seizure-free patients have a significantly higher perturbation strength (MNI: $\hat{p} = 1.40$ (3.95); CHUGA: $\hat{p} = 1.28$ (0.56)) compared to non-seizure-free patients (MNI: $\hat{p} = 1.14$ (0.36); CHUGA: $\hat{p} = 0.99$ (0.31)) with moderate effect. ~~In both figures, n~~Non-seizure-free and seizure-free patients are shown as red and green dots, respectively. MRI-negative patients are represented as a circle, and MRI-positive are presented as a square. **The center line of the boxplot represents the median, and the box limits represent the 25th and 75th percentile. The whiskers represent the complete range of values which are within 1.5 times the interquartile range.** **b** The AUC for classifying the implantation scheme via the patients’ surgical outcome is 0.75 (95% CI: 0.63,0.93). A threshold $\tilde{p}^* \cong 1.20$ was selected in the MNI cohort to achieve a specificity of 0.61 and sensitivity of 0.76, as shown in the ROC curve. **c** **The AUC on the CHUGA patient cohort is 0.80 (95% CI: 0.63,1.00).** \tilde{p}^* is tested on the CHUGA cohort, obtaining a sensitivity of 0.61 and specificity of 0.75. ~~Outliers were removed from the boxplot for better visualization of the CHUGA cohort results~~ **c** The ROC curve of the IED- γ spatial system in the CHUGA cohort. The threshold is visualized as a horizontal dotted line in **a**. **Summary statistics are reported as median (IQR).** Abbreviations: IED=interictal epileptiform discharge, IQR=interquartile range, MNI=Montreal Neurological Institute, CHUGA=Grenoble Alpes University Hospital Center, **CI=confidence interval**, ROC=receiver operating characteristics. Statistical significance is shown in asterisks: * $p < 0.05$, ** $p < 0.01$.”

Line 215,217: Referral to Figure 7 is inappropriate since the figure does not display the sensitivity and specificity of surgical outcome classification.

Response:

The reference is now removed. The figure is now referred to differently as follows:

On revised Manuscript highlight Page 12-13, Paragraph 3, Lines 251-255 under “Results”

Indeed, \hat{c}_1 and \hat{c}_2 are significantly different from each other (see **Supplementary Fig. S5 S8**). The scatter plots for MNI and CHUGA patient cohorts are shown in **Figures 7a and 7b**. The centroids were then used to classify surgical outcome, resulting in 64.7% sensitivity and 60.6% specificity in the MNI patient cohort (**Figure 7a**). The centroids were tested on the CHUGA patient cohort, resulting in a sensitivity of 55.6% and specificity of 75.0% (**Figure 7b**). Therefore, we hypothesized that the model could delineate good and poor implantations, with the \hat{c}_1 cluster classifying good implantations, and the \hat{c}_2 cluster classifying poor implantations.

Line 491: It is stated that new bipolar channel coordinates were computed as midpoints. Does the y-Axis on Figure 6 represents the bipolar channels or the original ones?

Response:

The y-axis represents the bipolar channels, given that a channel is represented by its perturbation strength and its distance to the region with high perturbation strength as described in **Methods** under “**constructing the SP map**”.

We made the following additions for more clarity.

On revised Manuscript highlight Page 11-12, Paragraph 4, Lines 228-231:

*“**Figures 6a and 6b** show clear examples of SP maps in two patients, P4 and P58, who had a good and poor implantation coverage, respectively. The y-axis of the SP map represents the bipolar channels, which are ranked by their perturbation strength in descending order. The x-axis represents the distances of each bipolar channel to the region of high perturbation strength. For patient P4, the implantation successfully identified a focal generator located...”*

Line 504: The terms “perturbation”, “spatial system”, and “response” are used interchangeably in the text. A clear explanation/statement about what these terms represent it would be helpful.

Response:

We apologize to the Reviewer for this ambiguity. We now clarified in the Methods section that (1) perturbation refers to the act of changing the system, (2) spatial system refers to the coupling of the interictal biomarker and the distance to the spatial reference, and (3) response refers to the measured change in the spatial system.

We added 2 more sentences for clarification in the revised Manuscript highlight Page 26, Paragraph 2, Lines 583-586

“The interictal biomarker used in this study is the IED- γ rate, as it demonstrated significant specificity to the EZ in our previous study¹². The overview of the framework is illustrated in **Figure 1**. In this study, a “spatial system” refers to the coupling of interictal biomarkers with their spatial distribution. A “perturbation” refers to the act of changing this spatial system. A “response” refers to the measured change in the spatial system when it undergoes some perturbation.”

Supplementary Figure S1: It is stated that out of 75 patients, 50 were eventually selected (40+10 in the flowchart), while 22 were excluded. Please clarify. Also, in the flowchart CHUGA has 42 patients while in text (line 441) there are 40 which are indicated for the CHIGA. Maybe add a subtraction symbol in the flowchart to make it meaningful. I am not sure whether I am missing something here.

Response:

We apologize for the confusion caused by the Figure legend and the main text. There was a typo in the total number of patients screened with 512 Hz sampling rate and follow-up > 1 years. We thank the Reviewer for their attention to detail, and have fixed this issue as follows:

On revised Manuscript highlight Page 23, Paragraph 2, Line 505 under “Methods”

We screened a total of 72 consecutive focal drug-resistant epilepsy patients from the MNI and ~~40~~ 37 consecutive patients from the CHUGA who underwent SEEG sampled at greater than 512 Hz with subsequent open resective surgery between 2009 and 2019; all patients had a postsurgical follow-up duration of ≥ 1 year.

On revised Manuscript highlight Page 23, Paragraph 3, Line 513 under “Methods”

“Out of the 72 patients from the MNI and ~~40~~ 37 from CHUGA, 50 and 26 patients were included into our analysis...”

On Supplementary Figure S1 legend:

The flowchart was fixed such that the arrows now point outwards, symbolizing that the patients are being removed from the screened population (as shown below):

Supplementary Figure S1: Patient selection flowchart. Out of the ~~75~~ 72 patients, 50 patients have been considered for analysis as per the inclusion and exclusion criteria. Twenty-two patients were excluded due to the following reasons: patients classified as Engel IB-IIA ($n = 15$); no available interictal segment as per the inclusion criteria ($n = 6$); and the absence of spontaneous seizures recorded during presurgical investigation ($n = 1$).

Answers to Reviewer 2's comment

This manuscript ventures into a refreshing new direction in the epilepsy literature, by proposing an interesting and well-developed model using electrophysiological features alone, whereby one may rate or 'score' the adequacy (or quality) of an SEEG implantation procedure. The model combines SEEG channel-specific interictal biomarkers (interictal epileptiform discharges with preceding gamma activity, IED- γ) and their exponential decay rates to construct a spatial manifold capturing the dynamics of the epileptic system. Subsequently, the authors assess the impact on network coupling by delivering a virtual perturbation [simulated via removal of the seizure-onset zone (SOZ) contacts], and use the strength of the perturbation effect to classify the quality of the implantation scheme for its sampling of the SOZ. The result is a spatial perturbation map that serves as a useful tool for clinicians and scientists to grade the quality of an SEEG study, and the model is validated through several permutations of data analysis linked to clinical outcomes, with nice descriptions of case examples.

The model uses the appropriate level and scale of patient data, collected across two well-established and renowned epilepsy centers, using stringent inclusion criteria and suitable processing methods. This Reviewer appreciated the decision to exclude a priori, the more ambiguous Engel IB-IIA group of patients. The demographics are comparable between the MNI and the CHUGA cohorts, suggesting homogeneity between the two patient populations. The SEEG-derived features are firmly established in the literature, and the manuscript presents a sound and rigorous mathematical/computational approach, bolstered by validation of the data using boot-strapping methods and supplementary information. The latter includes excellent and representative case examples, highlighting the clinical utility and impact of the study. Assumptions or data thresholds are appropriately validated, either quantitatively or through references made in the literature. Clinical experts were used at appropriate time points, such as using a blind reviewer to assess whether a resection was incomplete due to functional considerations. The authors made appropriate corrections to their data processing pipeline, including the correction for patients with insufficient resection, once again applying a more stringent limitation to their data. The integration of probability modelling and unsupervised clustering (i.e. machine learning) techniques adds further to the novelty of the project, and to the best of this Reviewer's knowledge, these are utilized in straight-forward fashion staying consistent with good practices in the literature. The figures are clear, legends are concise, and readability and overall presentation are excellent.

The signal processing and analytical methods used in the manuscript are comprehensive and appropriately employed, applying well-established channel-specific signal properties to a unique inception of a spatial manifold system, whose perturbation is useful in providing qualitative interpretation and classification of the overall SEEG implantation. This data-driven approach is equation-free and independent of other potentially confounding factors, built up from the

electrophysiological properties inherent to the channel recordings themselves. The analytical approach is nicely described, and easy to follow, in a step-wise fashion. The statistical analyses appear sound and are well-described. The data interpretation and conclusions drawn are validated by the results presented, in addition to the wealth of supplementary information provided.

The manuscript is clear, easily understood and nicely organized. In fact, the step-wise progression of analytics is relatively easy to follow. Terms are nicely explained throughout the manuscript. The context is well-developed with useful and appropriate references, which serve to validate the methodological decision-points in the analytical workflow of this project. The manuscript cements its inception and utility in the broader context of the world of SEEG and epileptogenic brain networks, and has real-world applications to the field of epilepsy and epilepsy surgery. I commend the authors on an excellent and well-composed manuscript here.

Response:

We would like to thank the Reviewer for their very positive feedback on our manuscript.

Suggested Improvements:

This Reviewer is appreciative for the chance to provide feedback, in the hopes of further improving the manuscript. Some of these suggestions are conceptual, and some are stylistic, as follows:

1) Page 3, Lines 45-46 – There is some discussion in the epilepsy community over the definition of the Epileptogenic Zone (EZ). Here, the authors employed one common definition, i.e. ‘the minimum amount of cortex needed to be removed to achieve seizure freedom.’ However, I would strongly recommend including the alternate, network-based definition from Talairach and Bancaud (who invented the SEEG procedure), whereby the EZ is defined as the network or region of earliest seizure organization and propagation. This would perhaps better underscore emphasis on ‘networks’ rather than a singular ‘focus,’ and is a better, more accurate historical definition to use for the analytical methods described in this manuscript, which seek to evaluate the quality of SEEG implantations. Perhaps a reference to the original work by Talairach and Bancaud might be fitting here, also.

Response:

We thank the Reviewer for this excellent suggestion. We agree that the definition of the EZ by Talairach and Bancaud works well with our methodology with uses spatial networks of the EZ. We now added the exact wording of the Reviewer to the introduction section of the manuscript:

On revised Manuscript highlight Page 4, Paragraph 1, Line 49-53, under “Introduction”:

“...to identify the epileptogenic zone (EZ), ~~defined as the minimum amount of cortex needed to be removed to achieve seizure freedom~~². However, in complex cases, these tools may not be sufficient, and stereotactic EEG (SEEG) electrodes need to be inserted into the brain to obtain a better spatial resolution³. **Bancaud and Talairach defined the EZ as the network or region of earliest seizure organization and propagation**⁴.”

Citation was updated to include Bancaud et al., 1965

2) Page 3, Line 49 – “... it is limited by the fact that only a restricted brain region can be explored...”

SEEG is an invasive methodology that allows for exploration of brain ‘networks,’ and is not confined to a restricted brain region in its sampling of those networks. Rather, to clarify this sentence, I would amend to something like ‘... it is limited by the constraint of implanting a finite number of electrodes ...’

Response:

We agree with the Reviewer’s suggestion. Thank you. We have modified the text accordingly:

On revised Manuscript highlight Page 4, Paragraph 1, Line 54-55 under “Introduction”:

“While SEEG can be extremely useful, it is limited by the ~~fact that only a restricted brain region can be explored;~~ **constraint of implanting a finite number of electrodes**...”

3) Page 3, Line 56 – The spelling of the word ‘miss-identification’ should be corrected to ‘misidentification.’

Response:

Thank you. Done. On revised Manuscript highlight Page 4, Paragraph 2, Line 62-63, under “Introduction”:

“...inability to properly sample the “true” SOZ¹⁰⁻¹² resulting in ~~miss-identification~~ **misidentification** of the EZ.”

4) Page 4, Lines 79-80 – The reference formatting for ‘interictal biomarkers’ needs correction here, I believe. Are the authors referring to refs 1 and 11? It is written as 111.

Response:

We apologize for the referencing issues. This issue was found in other parts of the article and is corrected (there was a corruption in the EndNote plug-in which seems to duplicate

the reference number due to some error). We now carefully checked all references one by one manually to avoid any errors.

5) Page 4, Line 88 – The spelling of ‘persued’ should be corrected, to ‘pursued.’

Response:

On revised Manuscript highlight Page 6, Paragraph 1, Line 107-108, under “Introduction”:

*“...originating from the non-invasive investigation and no surgery should be **persued** **pursued**.”*

6) Page 8, Lines 168-169 – It is interesting that no effect or difference was observed between MRI-positive and MRI-negative patients in the two cohorts. It is counterintuitive, in that one would expect a better SEEG implantation and subsequent increased perturbation effect following virtual removal of the SOZ, in patients with MRI-positive findings. Perhaps the authors could include a short statement to explain their observation and possible justification, later in the discussion. Also, did the study perhaps include patients who underwent morphometric analytical post-processing (MAP)? Just curious, and if so, it would be interesting to clarify this point further.

Response:

The Reviewer asked an excellent question which raises a very important issue about lesions and surgical outcomes. We agree that lesional cases typically have a higher chance of having good implantations and therefore, should have better surgical outcomes. Indeed, all patients at the Montreal Neurological Institute (MNI) undergo advanced image post-processing in the Bernasconi lab, however, this is not systematically performed in the Grenoble Alpes University Health Center (CHUGA). It should be noted that in the lesional cases treated at the MNI, MRI imaging was performed and carefully analyzed to minimize the need to undergo invasive EEG investigation. Therefore, most of the patients with SEEG data in this study are those with very complex epilepsy, which cannot be simply described by the lesion alone. In addition, a previous study from our group has shown that the presence of MRI lesions in our SEEG cohort could only classify surgical outcome with an accuracy of 0.5³. Another study from the literature has further shown that the MRI lesion does not necessarily correlate with surgical outcome⁴. If indeed, the lesion encompassed the primary organization of seizure activity, then its removal should result in seizure-freedom. Given that lesion does not indicate seizure-freedom in all cases, it is evident that sampling the lesion is not necessarily sufficient for seizure-freedom. At least in the MNI and CHUGA cohorts, we did not find a difference between presence of absence of MRI lesions as depicted in Figure 4.

Indeed, we would like to thank you for giving us the opportunity to clarify this point in the manuscript, and have done so by inserting a paragraph in the following lines:

On revised Manuscript highlight Page 20, Paragraph 2, Line 429-436, under “Discussion”:

“The reason for the poor perturbation profile of the ripple-rate spatial system could be due to the confound of physiological ripples, given large spatial variability across different brain regions^{26,27}.

In addition, we found that the results were not correlated with the presence of a lesion (i.e., MRI-positive) when correlating the perturbation strength with surgical outcome in the virtual-removal SP framework (see **Figure 4**). Although MRI-positive cases should have a higher likelihood of a good implantation, it has been shown that MRI lesions do not necessarily correlate with surgical outcome in temporal lobe epilepsy evaluated by bitemporal intracranial implantations²⁶. Indeed, while most of the implanted patients at the MNI underwent advanced MRI imaging analysis, previous findings from our group showed no difference in outcome in case of an identified MRI lesion²⁸.”

7) Page 11, Lines 228-229 – It is curious that a high fraction of patients (nearly 62%) were deemed as ‘incomplete resections.’ This seems high, and it would be useful for the authors to clarify how this was determined. Was the surgeon considering the resection as ‘incomplete’ immediately post-operatively (i.e. surgical goal not achieved); was the resection intentionally limited owing to overlapping functional networks (in which case, the surgical goal may still have been achieved, and considered ‘complete’); or is it based on recurrence of seizures, leading to repeat imaging, in which case the surgical target was not resected (again, ‘surgical goal not achieved’)? Some further clarification will be essential here.

Response:

Thank you for pointing this out. It was determined retrospectively by a neurologist not involved in the patients’ management and was based on the overlap between the identified SOZ channels in the SEEG and the surgical cavity as done in our previous studies^{3,5}.

We defined the SOZ as channels with first unambiguous ictal EEG changes independent of fast activity content and seizure onset pattern, as done in all of our previous papers^{2,5-7}. A complete resection means that all SOZ channels were included in the resected channels (therefore, they were located within the resection cavity), and an incomplete resection means that one or more channels were not included in the resected channels.

The reason we used these criteria was that in most of the older cases no written information was found in the neurosurgery documentation regarding why they removed or did not remove certain channels of the SOZ. Based on the anatomical location of the SOZ, we assume in most cases this was related to the overlap between the functional cortex and the SOZ channels. We agree that the wording of incomplete resection is ambiguous, as resections were indeed as complete as deemed necessary and feasible given relationship with functional cortex. In the revised version of our paper, we opted

now to label them as complete removal of all SOZ channels and incomplete removal of all SOZ channels. We also list the following explanations for the incomplete removal of SOZ channels as outlined above.

The following changes were made in the Manuscript to clarify this issue further:

On revised Manuscript highlight, Page 13, Paragraph 2, Lines 263-271 under “Results”:

To control for these variations, an epileptologist assessed ~~for~~ whether patients in the MNI cohort ~~whether a patient~~ had incomplete resections of the measured SOZ (see **Methods**) ~~due to functional considerations~~. Indeed, their implantation might have been successful; however, ~~the surgery was incomplete~~ the close proximity of the SOZ to functional cortex could have restricted the extent to which the measured SOZ could be removed, likely explaining the poor surgical outcome. As such, we will define resections which did not completely remove the measured SOZ as incomplete resections. To investigate this factor in the results, ~~Therefore~~, we checked how many of our patients were classified by the algorithm as well-sampled and a had poor outcome. We found ~~indeed~~ that 62% (8/13) of these patients had incomplete resections of the measured SOZ.

On revised Manuscript highlight, Page 35-36, Paragraph 2, Lines 781-788 under “Methods”:

“To correct for the heterogeneities in poor outcome patients, and report ‘true’ classification results (i.e., good outcome = good implantation, poor outcome = poor implantation), a clinical expert (blind to the results of this study) marked whether a patient in the MNI cohort had an incomplete resection due to functional considerations. An incomplete resection was defined as a resection which did not remove the entire measured SOZ, regardless of the surgical objective, since the goal of surgery may not always be to remove the whole SOZ but also to spare the overlapping functional cortex. Based on our definition, a patient was marked as having a complete resection if all SOZ channels were included in the resected channels (i.e. all SOZ channels were located within the resection cavity). Otherwise, if one or more channels were not included in the resected channels, the patient was marked as having an incomplete resection. Then, a data-driven approach was taken to ensure...”

8) Page 21, Line 465 – The reference (300) at the end of the sentence, should be corrected (30?).

Response:

We apologize for the erroneous referencing. We corrected this accordingly.

9) Page 39, Fig. 1a (right figure, brain illustration with electrode localization) – The schematic illustrates electrodes penetrating the brain, and clearly shows the concept of contact distances used in the analyses. However, these electrodes appear to be unconventional and may be exaggerated in placement, for the effect

of explaining the analysis. Do the surgeons at these centers implant orthogonally? Obliquely? The figure shows a rather haphazard electrode implantation, and so perhaps some clarification in the body of the text will be helpful. And importantly, if this is a true implantation schematic, the authors may disregard this comment, as surgical practices do vary for this technique.

Response:

We thank the Reviewer for their attention to detail, allowing us to improve the quality of our Figures. We agree that the illustration of the SEEG implantation is exaggerated and was only meant to illustrate how the distances were calculated and is not based on the data of a real patient.

*We have made the necessary modifications to **Figure 1** in the manuscript, which is now based on an implantation scheme of a real patient, and modified the sentence in the Results section to properly refer to the Figure.*

On revised Manuscript highlight Page 8, Paragraph 1, Lines 144-146:

*“The main premise for our framework is that we assume a *continuously decaying rate of the interictal biomarker with the distance to the region where this biomarker is maximum in case of a well implanted unifocal epilepsy.* Indeed, we see an example of this spatial coupling for a seizure-free patient ~~and a non-seizure-free patient in **Figure 1b**,~~ and how this coupling could change if we simulate a poor implantation of the SOZ using the same seizure-free patient (**Figure 1b**).”*

The Figure and the modified figure legend are shown below:

Figure 1: The spatial perturbation framework. **a** Inputs to the spatial perturbation framework. IED- γ are detected by finding IEDs with significant gamma activity preceding its onset as performed in Thomas, et al. 12 and Ren, et al. 21. SEEG channel coordinates are obtained by performing linear co-registration of post-implantation imaging with a template in normalized MNI space ⁴³. Distances are calculated between each channel and a spatial reference, ~~which we defined in this study,~~ as the channel with the maximum IED- γ rate. **b** The SP framework characterizes the implantation scheme by coupling IED- γ rates and their distance to the **spatial reference channel with maximum IED- γ rate**. Examples are shown in a seizure-free case patient, and a non-seizure-free case (right) the same patient, but three electrodes removed which are involved in the SOZ, to clearly simulate a poor implantation (i.e., **downsampled seizure-free**). **c** A perturbation was applied to the spatial system by virtually removing the **measured SOZ**. The change in the spatial coupling was quantified as the perturbation strength and **was is** used to classify the implantation scheme. **d** Another kind of perturbation was applied to measure the implantation quality without knowledge of the SOZ. ~~This was done by changing~~ **permuting the spatial reference used to compute the distances with each channel, and measuring T** the change in the spatial coupling (i.e., **perturbation strength**). ~~corresponding to~~

~~each channel used as the new spatial reference, was quantified, and~~ This is translated into the spatial perturbation map. ~~The translation involves~~ by ranking the channels by their perturbation strength (y-axis) and ~~ranking~~ their distances to the centroid of the area with high perturbation strength (x-axis). The color intensity shown on the SP map is the perturbation strength (z-axis).

Note: The SEEG implantation shown in **a** is based on the implantation scheme of P2. Only a subset of six of 15 electrodes within the plane are shown, and the contacts are enlarged for clarity. In **b**, the seizure-free patient is used to demonstrate a well-sampled patient, and the same patient was used to demonstrate a poorly sampled SOZ by removing three electrodes which sampled the SOZ. This simulates an electrode configuration which fails to sample the 'true' SOZ. The spatial reference was restricted to the SOZ as done in the virtual-removal SP framework. All brain figures are based on real patient data, and are simplified for demonstration purposes.

Abbreviations: IED=interictal epileptiform discharge, SOZ=seizure onset zone, SP=spatial perturbation.

Answers to Reviewer 3's comment

The code files were very well organized with clear instructions on how to install, run, and replicate results of the manuscript titled "A Spatial Perturbation Framework to Validate Implantation of the Epileptogenic Zone". The code also contained description of the script, inputs, and outputs as well as commented lines to help the users modify and implement their own tests. A readme file contained instructions to install and use the code which made it easy to install, run, and visualize results. A license agreement and pseudocode were also provided with the code. Two sample EEG data files were also included to test the code (one for well sampled case and one for poorly sampled case). A breakdown of the revision and some recommendations are listed below:

- The code is well written, well described and commented whenever necessary except in the four scripts (boxplot_gramm.m, plotFeatureSpace.m, Scatter_gramm.m, computeSpikeGamma.m) which lacked any description or comments. For completion and interpretability, these scripts need to be described and commented like the others.

Response:

We thank the Reviewer for their comment, as it provides us with the opportunity to further improve the accessibility and reproducibility of our research. We added documentation for the accessory functions as requested. The change logs for the GitHub repository can be found in the attached PDF. This file is also found in the revised submission and labelled as a 'Related Manuscript File'.

change_logs_code_1.
pdf

- The code was revised and tested (all functions and scripts were tested one by one). All codes ran smoothly without errors and generated the expected results of the manuscript and as described in the readme file.

Response:

We thank the Reviewer for reviewing our code and ensuring that it functions as intended.

- The methodology described in the manuscript was perfectly replicated by the code and the pseudocode describes the methodology well.

Response:

We thank the Reviewer for their careful assessment.

- The manuscript had some ambiguities in terms of steps (specifically line 636 – 666 “Constructing the SP Map” section on page 29 which was hard to understand or follow since the steps were not clearly described mathematically), but the code and the pseudocode helped clarify them all. The authors should consider enhancing that paragraph in the manuscript.

Response:

We thank the Reviewer for their comment. We agree that it is not quite clear and have prepared a flowchart diagram as requested by Reviewer #1, which we hope will add more clarity to the paragraph (shown below).

Supplementary Figure S11: Detailed flowchart of constructing the SP map. Firstly, a spatial system is constructed using the maximum IED- γ channel as the spatial reference (right column). This spatial system is perturbed by permuting the spatial reference with

each channel (left column). The perturbation strength is computed as the change in the spatial system after applying a permutation with channel i . This is done for all channels available in the SEEG implantation scheme, which produces a distribution of perturbation strengths associated with each channel. A second-step perturbation is applied by redefining the spatial reference as the region of high perturbation strength, defined as the mean of the channel coordinates with perturbation strengths larger than the 70th percentile of the distribution. This is used as the x -axis of the SP map. The channels are sorted in ascending order and plotted in the y -axis. A uniform bin size of 200 is used in the x -axis for uniformity across patients.

The figure is also referenced in the following paragraph:

On revised Manuscript highlight Page 33, Paragraph 2, Line 734-735 under “Methods”

“...spatially connect the perturbation indices in proximity to each other ⁴⁶. **A detailed flowchart on the construction of the SP map can be found in Supplementary Fig. S11.**”

- **Some typos in the comments within the code scripts were found, the authors may consider correcting them (such as line 127 in computeSPMap.m [“feautres”→ features] among others).**

Response:

We thank the Reviewer for their attention to detail. Indeed, correcting these issues will help improve the code’s quality and professionalism. The changes made can be found in the attached PDF GitHub commit logs. These files are also found in the revised submission and labelled as a ‘Related Manuscript File’.

change_logs_code_2.pdf

change_logs_code_3.pdf

- **The methodology seems to work with any feature extracted from iEEG. The authors may consider mentioning this fact in the discussion highlighting its use and limitations when used with other features if any.**

Response:

We thank the Reviewer for this excellent comment. Indeed, the method can be applied to any interictal biomarker of the EZ. One requirement though is that it must be specific to the EZ.

This was also requested by Reviewer #1 as well, and the results for the virtual-removal SP framework is shown for spatial systems constructing using IEDs and for ripples. We found that their performance is not as good as IED- γ rates, which makes sense given that these biomarkers are less specific to the EZ (IEDs delineate the irritative zone ⁸, and physiological ripples can reduce the specificity of ripples) ⁹. We did not use fast ripples,

since the sampling rate of the test set is only limited to 512 Hz. We have included the results in the Discussion section accordingly:

On revised Manuscript highlight Page 19-20, Paragraph 3, Lines 419-428, under “Discussion”:

“To strengthen the foundation of our framework on the use of IED- γ activity, we investigated the virtual-removal SP framework using more traditional interictal biomarkers such as IEDs and ripples (80-250Hz) (see **Supplementary Fig. S10**). In theory, the framework can be applied to any interictal marker with the requirement that it is specific to the EZ. We found that IEDs constructed a sub-optimal spatial system given that they are rather unspecific to the EZ¹⁶. Virtually removing the SOZ perturbs the IED spatial system less than in the IED- γ spatial system in both MNI and CHUGA patient cohorts. We also found that ripples perform extremely poorly in the virtual-removal SP framework. The reason for the poor perturbation profile of the ripple-rate spatial system could be due to the confound of physiological ripples, given large spatial variability across different brain regions^{26,27}.”

The figure in Supplementary is shown below for reference

Supplementary Figure S10: Perturbation strength of IED- γ predicts surgical outcome. The perturbation strength is computed for each patient as well as different interictal EZ

biomarkers (described in **Methods**). (a) IED rates achieve near significance ($\hat{\rho}_{SF}, \hat{\rho}_{non-SF} = 1.30 (0.37), 1.15 (0.29)$; $p = 0.054$; $d = 0.34$; $AUC = 0.67 (95\% CI: 0.51, 0.87)$), IED- γ rates achieve significance with a moderate effect ($\hat{\rho}_{SF}, \hat{\rho}_{non-SF} = 1.40 (3.95), 1.14 (0.36)$; $p < 0.01$; $d = 0.50$; $AUC = 0.75 (95\% CI: (0.62, 0.94))$), however, ripple rates do not produce an SOZ network that evaluates the measured SOZ ($\hat{\rho}_{SF}, \hat{\rho}_{non-SF} = 1.15 (0.35), 1.06 (0.61)$; $p = 0.98$; $d = 0.005$; $AUC = 0.50 (95\% CI: 0.32, 0.71)$). (b) The ROC curves of perturbation strengths using IEDs, IED- γ , and ripples. The threshold used to classify surgical outcome is obtained from the ROC curve of the IED- γ spatial system. (c) We found the same trends in the CHUGA patient cohort, with IED rates now achieving significance; ripples remained non-significant (**IED**: $\hat{\rho}_{SF}, \hat{\rho}_{non-SF} = 1.20 (0.35), 0.97 (0.29)$; $p < 0.05$; $d = 0.53$; $AUC = 0.76 (95\% CI: 0.56, 1.00)$, **IED- γ** : $\hat{\rho}_{SF}, \hat{\rho}_{non-SF} = 1.28 (0.56), 0.99 (0.31)$; $p < 0.05$; $d = 0.60$; $AUC = 0.80 (95\% CI: 0.62, 1.00)$, **Ripples**: $\hat{\rho}_{SF}, \hat{\rho}_{non-SF} = 1.19 (0.43), 1.03 (0.98)$; $p = 0.80$; $AUC = 0.47 (95\% CI: 0.11, 0.85)$). In both figures, non-SF and SF patients are shown as red and green dots, respectively. MRI-negative patients are represented as a circle, and MRI-positive are presented as a square. Summary statistics are represented as median (IQR). \tilde{p}^* is tested on the CHUGA cohort, obtaining a sensitivity of 0.61 and specificity of 0.75. (d) The ROC curves of the IED, IED- γ and ripple spatial systems in the CHUGA cohort. The threshold is visualized as a horizontal dotted line in (a) and (c). Abbreviations: IED=interictal epileptiform discharge, MNI=Montreal Neurological Institute, CHUGA=Grenoble Alpes University Hospital Center, IED=interictal epileptiform discharge, IQR=interquartile range, ROC=receiver operating characteristics. Statistical significance shown in asterisks: * $p < 0.05$, ** $p < 0.01$

- Overall, this work is a very well written, well coded, and very useful as a methodology to discriminate between well-sampled and poorly sampled iEEG implantations.

Response:

We thank the Reviewer for carefully reviewing the functionality, quality and readability of our code.

References

- 1 Shi, W. *et al.* Spike ripples localize the epileptogenic zone best: an international intracranial study. *Brain*, awae037 (2024).
- 2 Thomas, J. *et al.* A Subpopulation of Spikes Predicts Successful Epilepsy Surgery Outcome. *Ann Neurol* **93**, 522-535 (2023).
- 3 Klimes, P., Peter-Derex, L., Hall, J., Dubeau, F. & Frauscher, B. Spatio-temporal spike dynamics predict surgical outcome in adult focal epilepsy. *Clin Neurophysiol* **134**, 88-99 (2022).
- 4 Massot-Tarrús, A. *et al.* Outcome of temporal lobe epilepsy surgery evaluated with bitemporal intracranial electrode recordings. *Epilepsy Research* **127**, 324-330 (2016).
- 5 Oderiz, C. C. *et al.* Association of cortical stimulation–induced seizure with surgical outcome in patients with focal drug-resistant epilepsy. *JAMA neurology* **76**, 1070-1078 (2019).
- 6 Astner-Rohracher, A. *et al.* Development and validation of the 5-SENSE score to predict focality of the seizure-onset zone as assessed by stereoelectroencephalography. *JAMA neurology* **79**, 70-79 (2022).
- 7 Frauscher, B. *et al.* Facilitation of epileptic activity during sleep is mediated by high amplitude slow waves. *Brain* **138**, 1629-1641 (2015).
- 8 Bartolomei, F. *et al.* What is the concordance between the seizure onset zone and the irritative zone? A SEEG quantified study. *Clinical Neurophysiology* **127**, 1157-1162 (2016).
- 9 Bruder, J. C. *et al.* Physiological ripples associated with sleep spindles can be identified in patients with refractory epilepsy beyond mesio-temporal structures. *Frontiers in Neurology* **12**, 612293 (2021).

REVIEWERS' COMMENTS

Reviewer #1 (Remarks to the Author):

The authors have addressed all my concerns.

Christos Papadelis, PhD

Reviewer #2 (Remarks to the Author):

I am satisfied with the new (and substantial) revisions for this manuscript, and recommend publication.

Thank you,
Demitre Serletis, MD PhD FRCSC FAANS FAES
Epilepsy Center, Neurological Institute, Cleveland Clinic, USA

Reviewer #3 (Remarks on code availability):

The authors addressed and satisfied all the concerns raised in the first review regarding the code. The authors also amended the methodology section to clarify the workflow of the methodology. I recommend this paper for publication.